# RECYCLED ATTENTION: EFFICIENT INFERENCE FOR LONG-CONTEXT LANGUAGE MODELS

## ABSTRACT

Processing long-context input imposes a heavy computational burden when deploying large language models. Recently proposed inference-time methods accelerate generation by attending only to local context. Despite its efficiency gains, this approach fails to capture all relevant information in the input, showing substantial performance drop in long-context benchmarks. We propose *recycled attention*, an efficient and effective method which alternates between full context attention and attention over a subset of input tokens. When performing partial attention, we leverage the attention pattern of a nearby token that has performed full attention and attend only to the top K most attended tokens. We evaluate our methods on RULER, a suite of tasks designed to comprehensively evaluate long-context abilities, and long-context language modeling tasks. Applying our inference method to off-the-shelf LLMs achieves comparable speedup to baselines which only consider local context while improving the performance by 2x. We further experiment with continued pre-training the model with recycled attention to improve the performance-efficiency trade-off.

## 1 INTRODUCTION

Recent large language models (LLMs) are trained to ingest extremely long inputs and generate long outputs (Meta, 2024; Gemini, 2024) to support a wide range of applications. However, deploying such long-context LLMs can be very costly. As the context length increases, LLMs see a linear increase in memory to store the Key-Value (KV) cache and a quadratic increase in time for attention computation. These two factors lead to high latency during inference; Adnan et al. (2024) showed that as context length increased 16x for the MPT-7B model (MosaicML, 2023), the inference latency increased by 50x, where 40% of the increase was due to the data movement of the KV cache.

To improve efficiency, prior work put a limitation on the size of KV cache, i.e. the number of past tokens that are available at each generation step. This leads to a meaningful gain along two axes: memory requirement and time for attention computation. To form a smaller KV cache, they make a locality assumption, only keeping most recent input tokens (Beltagy et al., 2020; Child et al., 2019) along with a fixed number of globally available initial tokens (i.e., StreamingLLM (Xiao et al., 2023)). Another line of work (e.g., $H_2O$ (Zhang et al., 2024), Keyformer (Adnan et al., 2024)) maintains a dynamically constructed fixed sized KV cache by identifying *key* past tokens from observed attention patterns and dynamically evicting the rest during generation.

These approaches reported little degradation in perplexity-based evaluation for the next token prediction task. However, they show a significant drop in performance (Sun et al., 2024) on long-context benchmarks that require synthesizing information from non-local contexts (Hsieh et al., 2024). For example, on the simple needle-in-a-haystack (NIAH) task, both StreamingLLM (Xiao et al., 2023) and $H_2O$ (Zhang et al., 2024) report less than 8% accuracy compared to 100% for vanilla attention. Keeping a smaller KV cache is problematic when LLMs is tasked with synthesizing information from long context, going beyond next token prediction where local contexts suffice. Once a key input token is eliminated from the KV cache (either through locality assumption or by eviction during the generation process), there is no way to recover access to the eliminated token. When LLMs are tasked to generate long text, it gets harder to predict which input tokens are useful in advance.

In this work, we propose a novel approach, Recycled Attention, that focuses on reducing inference time while comprehensively capturing long-context inputs. We keep the full KV cache throughout

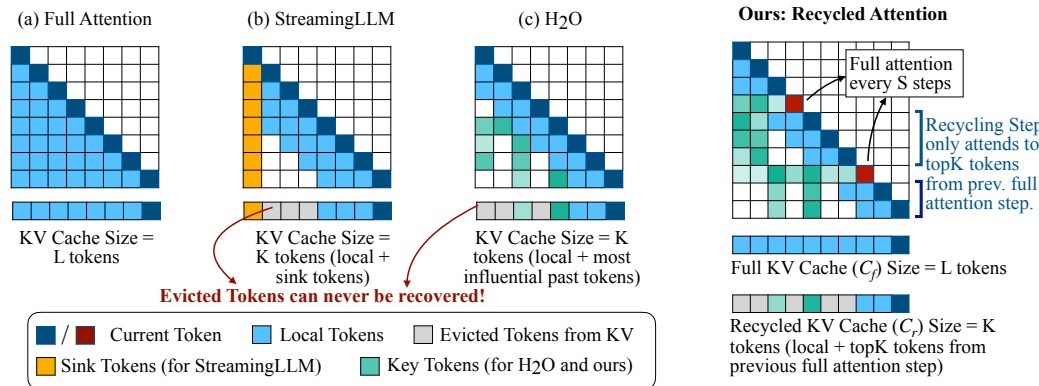

Figure 1: Illustration of our Recycled Attention method (right) compared to baselines (left). Our approach alternates between full attention steps (i.e. over all past tokens) and *recycled* attention steps (i.e. over a reduced KV cache of *key* tokens) during generation. By restricting the full attention computation to once in every S steps, Recycled Attention is able to achieve comparable speedups to baseline models with smaller KV without degrading performance on long-context benchmarks.

the inference (thus no gain in memory footprint), but perform attention over a dynamically constructed smaller KV cache, retaining gain in inference speedups. Our method flexibly alternates between two modes of generation: generation that involves an attention over the full KV cache and generation that computes an attention over a subset of tokens (see Figure 1). We choose this subset of tokens by taking top K attended tokens from the most recent generation step involving attention over the full KV cache (thus the term *recycling* attention). In this work, we have a fixed strategy for alternation: full attention every $S$ steps and recycled attention for the next $S - 1$ steps. Our design choices is supported by the analysis that neighboring tokens during generation place high attention mass over a similar subset of past tokens. Our work (no KV eviction, dynamically constructed smaller KV) establishes a middle ground between full attention (no KV eviction, high latency, high performance) and sparse attention (KV eviction, reduced latency, low performance).

We evaluate our approach in language modeling task and RULER (Hsieh et al., 2024) benchmark, a suite of tasks designed to evaluate long-context models, as well as datasets from LongBench(Bai et al., 2023). Applying our inference method to two off-the-shelf LLMs (Meta, 2024; Yang et al., 2024a) achieves comparable speedup to prior work with limited KV cache while improving the performance on long-context benchmark by 2x. We further experiment to continued pre-training the model with recycled attention, bringing further gains. To summarize, our contributions are

- We propose *recycled attention*, an inference-time method to accelerate generation with long input.
- We comprehensively evaluate our methods to two long-context models and a suite of long-context tasks, including downstream tasks and language modelling tasks. We find that our method achieves up to 2x wall clock time speedup while preserving performance, especially on downstream tasks which require access to information throughout the input.
- We investigate further improvements: continued pre-training LLM with recycled attention and deciding when to perform full attention based on query similarity.

## 2 RECYCLED ATTENTION FOR LONG-CONTEXT LLMS

### 2.1 PROBLEM SETTING AND NOTATION

Let $M$ be a language model trained to estimate the conditional probability of all output sequences given an input $x$. At inference time, $M$ generates an output $\hat{y} \sim M(x)$ in two steps: (1) **Pre-filling stage**: $M$ ingests the input $x = x_1, \cdots x_L$ and stores the KV cache for all $L$ tokens across all layers of the transformer model, and (2) **Generation stage**: generate one token $y_i$ at a time from the conditional distribution $P_M(y_i|x, y_1 \cdots y_{i-1})$. At each step, the model attends to the KV cache of all previous tokens, and also updates the KV cache to include the current token's key-value pairs.

**Our goal is to reduce the inference latency during this second stage of the generation process.** There are two main factors that contribute to this increased latency; first, the attention computation increases quadratically with input length $L$. Second, a large $L$ necessitates maintaining a large KV cache of past tokens, and 40% of the inference latency can be attributed to the data movement of this large KV cache from the GPU HBM (Adnan et al., 2024).

Prior approaches (Xiao et al., 2023; Zhang et al., 2024; Adnan et al., 2024) achieve inference time speed-ups by limiting the size of the KV cache to a fixed size $K$. StreamingLLM (Xiao et al., 2023) constructs this fixed size KV cache by retaining only the initial and recent tokens (illustrated in Figure 1(b)) while $H_2O$ (Zhang et al., 2024) retain a mix of local tokens and *key* past tokens dynamically identified during generation (see Figure 1(c)). For both approaches, once tokens are evicted from the KV cache, they cannot be recovered in subsequent generation steps. This can be particularly catastrophic in long-context scenarios where key tokens are challenging to identify in advance, e.g. cases where instructions inquiring about the past tokens are located towards the end of the input. Consequently, methods that evict tokens from the KV cache often report poor performance on benchmarks like RULER (Hsieh et al., 2024) that are designed to test reasoning and information synthesis capabilities over long-contexts.

Instead of permanently evicting tokens for all future steps, we ask: **can we distinguish between important and unimportant tokens for the attention computation for the next S time steps?** Our key hypothesis is that consecutive tokens in a sequence likely place the majority of the attention weights over a similar subset of tokens in the context, and this can be leveraged to increased inference efficiency. We test this hypothesis for the LLaMA-3.1-8B model in the subsection below.

## 2.2 ATTENTION MASS OVERLAP BETWEEN NEIGHBORING TOKENS

**Setting:** We randomly sample five examples from the Arxiv split of the RedPajama dataset (Together, 2023) and compute the attention weights over past tokens for all layers and all time steps. Next, for time step $t = 8K$, we identify the topK($= 1024$) past tokens based on attention weights independently for each layer. Then, for subsequent attention computations for tokens at times steps $t = 8K + i$, varying $i$ from 1 to 10, we compute the fraction of the attention mass placed on $t = 8K$'s topK tokens. Figure 2 shows this attention recovery rate for different step $i$ from the full attention step, averaged across all layers of the transformer model (shown in blue). The graph clearly demonstrates that the topK tokens at $t = 8K$ include the past tokens that contribute, on average, more than 90% of the attention weights at subsequent times $t = 8K + i$. Based on this observation, our key idea is to alternate between full attention over the entire KV cache of past tokens every $S$ steps, and a more time-efficient attention over only K tokens for the next $S - 1$ steps, where these K tokens are selected to be the highest weighted tokens during the previous full attention step. We call this strategy Recycled Attention, as we recycle the topK tokens from a previous time step in lieu of full attention.

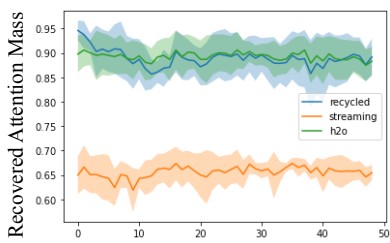

Figure 2: Fraction of the total attention mass recovered at time $t = T + i$ by the topK past tokens in the KV cache, where these topK tokens are selected based on attention scores at $t = T$. Compared to StreamingLLM, topK tokens recover a larger fraction of the total attention mass.

Note that the graph in Figure 2 also reports the fraction of the full attention weight placed on tokens corresponding to StreamingLLM's cache of similar size K (shown in orange), comprising of the initial "sink" tokens and the local tokens (see Figure 1b for KV construction strategy). Compared to our proposed strategy, StreamingLLM reports a much lower attention mass recovery rate ($\sim 0.65$ compared to $0.9$ for our approach) and is consequently worse at approximating full attention.

## 2.3 METHODOLOGY AND IMPLEMENTATION

Given a language model $M$ and a sequence of input tokens $x_1, ..., x_L$, we present the pseudocode for generating the output sequence of tokens using Recycled Attention in Figure 3.

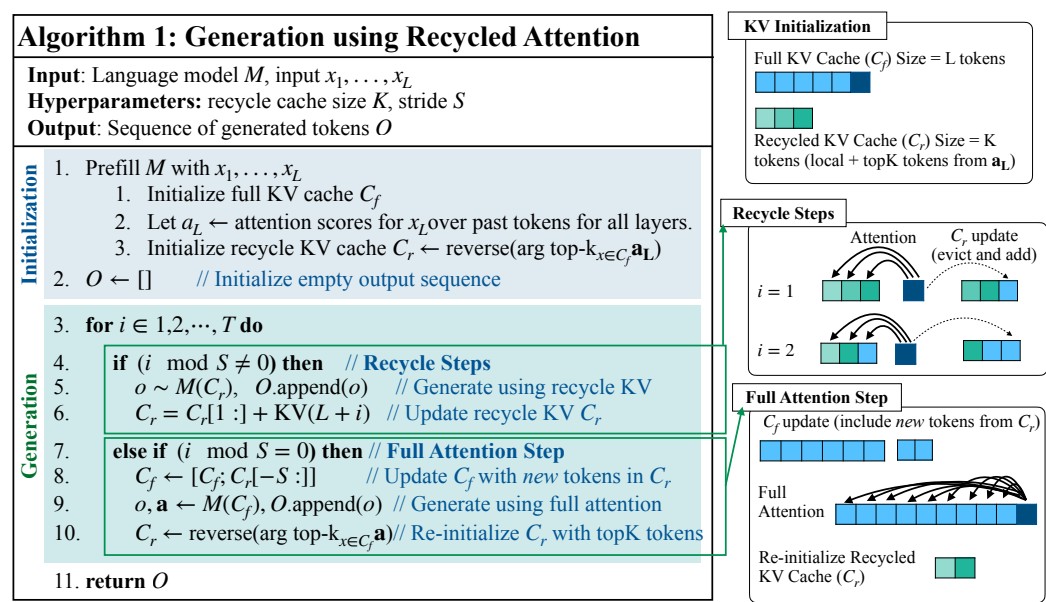

Figure 3: Pseudocode for Recycled Attention. We use $M(C)$ to denote performing a forward pass with the language model while computing attention over the key-value pairs in cache $C$.

Our approach maintains two separate KV caches $C_f$ and $C_r$ (size $\propto K$), corresponding to the full and recycle attention steps respectively. Given input $x_1, ..., x_L$, we first prefill $M$ using the vanilla full attention computation and initialize our full KV cache $C_f$ with the first $L$ tokens. We also obtain the attention scores $\mathbf{a_L}$ for the last token $x_L$ at each layer of model $M$. We initialize our recycle KV cache $C_r$ with the key-value pairs of the topK tokens at each layer based on attention scores.

At each recycle steps, i.e. $S-1$ contiguous steps after every full attention step, we generate the next token $y_t \sim M(C_r)$ using the smaller recycled KV cache $C_r$ to compute attention. This leads to a reduction in both the attention computation FLOPs as well as the latency due to movement of KV cache (we only move the smaller KV $C_r$ instead of the larger full KV $C_f$, where $|C_r| << |C_f|$). As $C_r$ is updated with the KV cache of the new input tokens (i.e. the generated token from previous step) in the forward pass, we remove the recycled token which received the lowest attention score from $C_r$ to maintain a fixed size.

At each full attention step that occurs every $S$ steps, we first update the $C_f$ KV cache with the key-value pairs of the $S-1$ tokens from the recycle step. Next, we generate the next token $y_t \sim M(C_f)$ using the full KV cache $C_f$. Finally, we follow the same procedure as above to reset the recycle cache $C_r$ with the topK tokens from each layer of the current time step.

**Compatibility with Flash Attention**   FlashAttention (Dao, 2024) improves standard attention computation on GPU by reducing data movement, significantly improving the memory and speed efficiency. It achieves this by directly producing the output for the attention blocks, without storing the $O(L^2)$ attention matrix. However, we rely on these attention scores to select the topK tokens during the full attention steps and construct our recycled KV cache $C_r$ (lines 9-10 of Algorithm 3). To make our method compatible with FlashAttention, we implement an extra step to re-compute the attention score when we perform full attention. As we only perform full attention at stride of $S$, this does not introduce significant overhead. Additionally, note that other methods that use attention patterns (e.g. $H_2O$) will also show reduced speed-gain when using in conjunction with FlashAttention.

**Memory and time requirements**   Table 1 shows a comparison of the memory and attention compute requirements for our Recycle Attention method and baseline approaches. Recycle Attention uses a similarly sized KV cache memory compared to vanilla attention ($L+K$ vs $L$, where $K << L$) but larger than efficient KV strategies like StreamingLLM and $H_2O$. Our method substantially re-

Table 1: Comparison of the time and memory requirements of Recycle Attention and baseline approaches. Suppose an LLM ingests a sequence of $L$ input tokens and generates $T$ output tokens. We report the memory requirement for storing the input KV cache and time required to generate the $T$ output tokens. Let $K$ denote the size of the reduced KV cache of baseline non-vanilla approaches. We use the same size $K$ for our recycle KV cache in Recycle Attention and use $S$ to denote the stride. Additionally, we report NIAH performance measured for the Llama-3.1-8B model, with $K = 4096$ and $L = 32,768$ for all approaches.

| | Vanilla | H$_2$O | StreamingLLM | SnapKV | Recycled Attention (Ours) |
|---|---|---|---|---|---|
| **Memory** | $L$ | $K$ | $K$ | $K$ | $L + K$ |
| **Time** | $T \times L$ | $T \times K$ | $T \times K$ | $T \times (K + T)$ | $T \times \frac{L}{S} + T \times K$ |
| **NIAH Accuracy** | 100 | 8 | 7 | 77 | 98 |

duces the time for the attention computation compared to vanilla attention by setting the recycle KV size $K << L$. Our strategy also allows us to remain performant on tasks such as NIAH compared to vanilla attention, in contrast with other KV eviction strategies. We provide a detailed comparison of wallclock times and performances on various tasks against baslines in Section 3.

## 3 EXPERIMENTAL SETTINGS

We evaluate our method on two long-context language models **Llama-3.1-8b** (Meta, 2024) and **Qwen-2-7b** (Yang et al., 2024a). **Llama-3.1** is pre-trained with 8K tokens, followed by a continued pre-training stage to increase the context window to 128K. **Qwen-2** is continued pre-trained with up to 32k tokens, and adopted YARN (Peng et al., 2024) and Dual Chunk Attention (An et al., 2024) to enable processing of up to 128k tokens. As both models employ Grouped Query Attention (Ainslie et al., 2023), we use a single aggregated attention score for all query heads (max over all query heads) in the same group to identify the top K tokens.[1]

### 3.1 TASKS

We evaluate our approach on language modeling and a suite of downstream proxy tasks for long-context evaluation (Hsieh et al., 2024). For both tasks, report the task performance and inference speed measured by wall clock time.

**Downstream tasks** We test our method on RULER (Hsieh et al., 2024), a suite of tasks designed to evaluate long-context models. It includes tasks that require retrieval capabilities (e.g. Needle-in-a-Haystack) as well as those that require aggregating information over the long context. We follow Hsieh et al. (2024) and evaluate our methods on 13 tasks from four categories of RULER. We evaluate on context length of 32K and 64K, with 100 examples for each {task, context length}. We additionally report on two tasks from LongBench in Section A.2 in the Appendix.

**Language Modeling** We evaluate language modeling perplexity on the Arxiv and Book split of RedPajama (Together, 2023), and PG19 (Rae et al., 2019). We evaluate on 16k and 100k context size for the two respectively. We report results on 100 sequences for each domain. Following prior work (Yen et al., 2024b), we report the perplexity on the last 256 tokens of each sequence.

### 3.2 BASELINES

We compare Recycle Attention against the following baselines: (1) **Vanilla** attention baseline which uses the entire KV cache to generate tokens. (2) **StreamingLLM** (Xiao et al., 2023) inferences by attending to a KV cache consisting of "sink tokens" and recent tokens, discarding all other tokens. Following previous work, we maintain a cache with 4 sink tokens and $K$ - 4 recent tokens. (3)

---

[1]Our ablations show that taking the max outperforms other aggregation method such as mean, or relying solely on one of the query head in the group. We detail this more in Table 7 in the Appendix.

Table 2: Performance on the RULER benchmark for LLama-3.1-8B and Qwen-2-7B. The results show that Recycled Attention achieves a comparable speedup to prior approaches, while substantially outperforming them based on accuracy across all settings.

| Method | stride | K | LLama-3.1 | | | | QWEN-2 | | | |
| | | | 32K | | 64K | | 32K | | 64K | |
| | | | Acc ↑ | time(s) ↓ | Acc ↑ | time(s) ↓ | Acc ↑ | time(s) ↓ | Acc ↑ | time(s) ↓ |
|---|---|---|---|---|---|---|---|---|---|---|
| Vanilla | - | - | 90 | 1.71 | 82 | 2.40 | 79 | 2.55 | 57 | 4.93 |
| $H_2O$ | - | 4096 | 21 | 2.15 | 11 | 2.29 | 16 | 1.94 | 11 | 1.94 |
| StreamingLLM | - | 4096 | 22 | **1.23** | 17 | **1.21** | 21 | **1.17** | 11 | **1.19** |
| StreamingLLM++ | 50 | 4096 | 22 | 1.25 | 17 | 1.33 | 21 | 1.21 | 11 | 1.29 |
| SnapKV (kernel=7, w=32) | - | 4096 | 72 | 1.64 | 62 | 1.73 | 57 | 1.43 | **31** | 1.60 |
| Recycled | 50 | 4096 | 63 | 1.27 | 50 | 1.29 | 32 | 1.21 | 20 | 1.20 |
| Recycled (kernel=7) | 50 | 4096 | **79** | 1.26 | **65** | 1.29 | **58** | 1.20 | **31** | 1.20 |

**StreamingLLM++**: we also implement a modified version of StreamingLLM that is equivalent to our Recycled Attention method in terms of both computation and memory requirements. Similar to our approach, StreamingLLM++ performs full attention at a stride $S$, i.e. every $S$ steps, to match the attention operations of Recycle Attention. (4) $H_2O$ (Zhang et al., 2023) maintains a KV cache which contains recent tokens and "heavy hitters", defined by high cumulative attention scores. We set the heavy hitter size and recent cache size to be $K/2$. (5) SnapKV (Li et al., 2024) considers the average attention scores of the last few tokens ("observation window") in the prompt to decide the KV cache to keep. It further applies max pooling over consecutive tokens' attention score, instead of relying on the token's attention score, to decide token to keep. We set the observation window size to 32 and kernel size to 7 following Li et al. (2024).[2]

## 3.3 INFERENCE SETTINGS

We prefill the model with the input and measure wall clock time for the generation phase for each of the method. We generate 50 tokens for the RULER tasks and 256 tokens for the language modeling task. We perform our experiments on 1 A100 80GB GPU with Flash Attention (Dao, 2024). We report a fixed set of $K$ and $S$ for the tasks, and perform ablation study on varying these two hyperparameters in Section A.1 in the Appendix. We include a variant of Recycled Attention with max pooling with kernel size=7 to cluster KV cache, same as SnapKV.

## 4 RESULTS

**RULER** For this set of experiments, we set $K = 4096$ for all baseline models, where applicable. For Recycled attention and Streaming++, we set stride $S = 50$.

Aggregate accuracy results and the generation time per example for the RULER benchmark are reported in Table 2. All methods aiming to achieve inference speedup by evicting tokens from the KV cache permanently (e.g. StreamingLLM, $H_2O$) shows substantial degradation. **Recycled attention significantly outperforms other non-vanilla approaches by over 2x in terms of aggregate accuracy**. However, we note that the performance degrades substantially compared to vanilla attention. In terms of speed-up, our method achieves similar speedup to StreamingLLM/StreamingLLM+, followed by $H_2O$ model. As input context length increases, inference time for vanilla method scales linearly, while the other methods' inference time remain at the same ballpark with a fixed $K$.

Note that $H_2O$ relies on calculating attention scores at each time step to identify the "heavy hitter" tokens which FlashAttention does not store. Thus, the inference speed-up is not as significant in certain setting (with 1/8 KV cache size for 32K input) when used with Flash Attention. For our Recycled Attention approach, we only explicitly re-compute the attention score every $S$ steps, which do not introduce as heavy an overhead.

---

[2]Our proposed method of constructing the top K cache is equivalent to SnapKV with a window size of 1 and kernel size of 1. While it is not feasible to apply a window size greater than 1 for our method as it will require access to attention scores at *each* decoding step, it is possible to apply the kernel method to our approach.

Table 3: Per-task performance of Llama-3.1-8B on RULER subtasks. For non-vanilla methods, we set the $K = 4096$.

| Method | niah_single | multi_key | multi_query | multi_value | fwe | vt | cwe | qa |
|---|---|---|---|---|---|---|---|---|
| *Context size: 32K* | | | | | | | | |
| Vanilla | 100 | 98 | 99 | 99 | 93 | 99 | 65 | 61 |
| $H_2O$ | 7 | 7 | 6 | 6 | 78 | 38 | 39 | 34 |
| Streaming | 8 | 13 | 13 | 13 | 93 | 12 | 4 | 42 |
| StreamingLLM++ | 8 | 13 | 13 | 13 | 93 | 10 | 5 | 43 |
| SnapKV (kernel=7, w=32) | 77 | 68 | 99 | 98 | 83 | 99 | 56 | 61 |
| Recycled | 98 | 35 | 59 | 37 | 90 | 99 | 20 | 59 |
| Recycled (kernel=7) | 99 | 60 | 98 | 99 | 83 | 99 | 44 | 63 |
| *Context size: 64K* | | | | | | | | |
| Vanilla | 100 | 90 | 96 | 99 | 91 | 98 | 3 | 54 |
| $H_2O$ | 3 | 2 | 2 | 4 | 52 | 3 | 8 | 20 |
| Streaming | 8 | 7 | 7 | 8 | 90 | 5 | 0 | 33 |
| StreamingLLM++ | 8 | 7 | 7 | 8 | 90 | 5 | 0 | 35 |
| SnapKV (kernel=7, w=32) | 74 | 42 | 90 | 88 | 71 | 92 | 5 | 48 |
| Recycled | 80 | 30 | 26 | 17 | 79 | 95 | 3 | 51 |
| Recycled (kernel=7) | 96 | 41 | 85 | 84 | 72 | 93 | 4 | 49 |

As RULER consists of a diverse range of task, we report per-task fine-grained performance for Llama-3.1 in Table 3. Recycled attention performs the best at tasks that require retrieving a piece of information in the context (including Needle-in-a-haystack (NIAH), Question Answering (QA) and Variable Tracking (VT)). $H_2O$ and StreamingLLM suffers at these tasks as the information falls out of the KV cache. However, recycled attention does not perform well for task that requires aggregating the information in the context, such as frequent word extraction (fwe), lagging behind $H_2O$ and StreamingLLM. Performances for tasks which requires retrieving for multiple pieces of information (multiple keys or multiple queries) are worse compared to the task with (single key, single value) when using the same $K$. We later show in ablation study in Section A.1 increasing the size of $K$ leads to improvements in such tasks.

**Language Modeling** For context size 16K, we fix $K = 2048$ and $S = 10$ both LLaMA-3.1 and QWEN-2. For context size $100K$, we report results using $K = 2048$ and $32,768$, and $S = 256$.

Table 4 outlines the perplexity-based performance of the baselines and recycled attention approach. For LLaMA-3.1, recycled attention achieves better perplexity and comparable inference speeds compared to StreamingLLM when the KV cache size is 1/8 of a 16K context. This shows that the model benefits from attending to tokens outside of local context window. We observe that $H_2O$ outperforms our approach by a small margin, but at the cost of a substantially higher inference time per example (10.77 for $H_2O$ vs 6.05 for our method). Overall, **our recycled attention approach achieves a better trade-off between inference speeds and task accuracy compared to non-vanilla approaches for both LLaMA-3.1 and QWEN-2 models** for the 16K context size setting.

When we scale up the context length to 100K, we find differing trends between the LLaMA-3.1 and QWEN-2 models. For LLaMA-3.1, we observe that recycled attention reports better perplexity but worse inference speeds compared to non-vanilla baseline methods. However, baseline approaches outperform recycle attention for the QWEN-2 model. We analyze the attention pattern to investigate this in Section A.1.

## 5 CONTINUED PRE-TRAINING WITH RECYCLED ATTENTION

So far, we use the off-the-shelf LLMs as is, only modifying the inference method. This creates a discrepancy between model training and inference assuming LLM is trained with vanilla full attention setting. In this section, we experiment with continued pre-training the model with recycled

Table 4: Perplexity results on language modeling task for LLama-3.1-8B and QWEN-2-7B. We report performances for Arxiv (the first number) and Book (the second number) and PG19.

| Method | K | Stride | LLama-3.1-8B | | QWEN-2-7B | |
|---|---|---|---|---|---|---|
| | | | time(s) ↓ | PPL ↓ | time(s) ↓ | PPL ↓ |
| *Context size: 16 K (Arxiv and Book)* | | | | | | |
| Vanilla | - | - | 7.63 | 2.22 / 7.07 | 8.85 | 2.33 / 8.26 |
| $H_2O$ | 2048 | - | 10.77 | 2.48 / 7.60 | 11.57 | 2.68 / 9.02 |
| StreamingLLM | 2048 | - | **6.92** | 2.62 / 7.94 | **5.71** | 2.75 / 9.10 |
| StreamingLLM++ | 2048 | 10 | 7.21 | 2.59 / 7.88 | 6.08 | 2.71 / 9.05 |
| SnapKV (kernel=7, w=32) | 2048 | - | 7.77 | 2.48 / 7.68 | 6.90 | 2.65 / 8.97 |
| Recycled | 2048 | 10 | 7.14 | 2.36 / 7.49 | 6.33 | 2.57 / 9.01 |
| Recycled (kernel=7) | 2048 | 10 | 7.14 | **2.32 / 7.40** | 6.33 | **2.47 / 8.73** |
| *Context size: 100 K (PG19)* | | | | | | |
| Vanilla | - | - | 18.11 | 8.24 | 40.42 | 13.28 |
| $H_2O$ | 2048 | - | 10.56 | 17.04 | 9.96 | **13.39** |
| StreamingLLM | 2048 | - | **5.94** | 9.53 | **5.72** | 13.58 |
| StreamingLLM++ | 2048 | 256 | 6.04 | 9.53 | 5.92 | 13.58 |
| Recycled | 2048 | 256 | 6.10 | **9.31** | 5.90 | 14.90 |
| $H_2O$ | 32,768 | - | 26.89 | 8.63 | 23.55 | 13.36 |
| StreamingLLM | 32,768 | - | **13.38** | 8.55 | **15.81** | **12.31** |
| StreamingLLM++ | 32,768 | 256 | 13.43 | 8.55 | 15.87 | 12.32 |
| Recycled | 32,768 | 256 | 13.52 | 8.46 | 15.89 | 13.50 |

attention, with the goal of teaching the models to adapt to attending over discontinuous tokens in the recycled cache.

**Data** We sample 200k data from the Arxiv split of RedPajama dataset[3] and filter out sequences with less than 8192 tokens. We split the data into 80%, 10% and 10% train/dev/test splits, resulting in 120k training data.

**Training** We train the model to adapt to recycled attention when predicting a sequence with 8142 with a prefilling length of 8092, $K = 2048$ and a stride of 50. Concretely, for the last 50 tokens $t_i$ in the sequence, attention is calculated over the 2048 tokens that received the highest attention score according to $t_{8092}$, as well as $\{t_{8093}, ... t_i\}$. For $\{t_0, ... t_{8092}\}$, attention is calculated with regular causal mask. We train the model with next token prediction loss for all the tokens in the sequence. We report implementation details in Section A.3 in the Appendix.

**Comparison sytstems** We compare fine-tuning with other inference methods (Vanilla, StreamingLLM, StreamingLLM++). For each method, we report the base performance from the pre-trained checkpoint (Base) and the performance after continued fine-tuning (+CPT).

**Results** We report the results of continued pre-training in Table 5 for both the language modelling and the 14 RULER tasks. We see that continued pre-training does not improve performance with vanilla inference method, likely as the model is highly optimized in this setting and trained with this data. We also observe very little performance gain through continued pre-training in other inference methods (StreamingLLM, StreamingLLM++). Yet, with Recycled Attention, we see a meaningful gain from continued pre-training in two stride setting (25, 50). Continued pre-training achieves a lower perplexity and higher accuracy with higher stride (50) compared to base model with a smaller stride (25), leading to a better performance-efficiency trade-off.

# 6 NEW SECTION: DYNAMIC STRIDE

Our experiments in Section 3 employs a fixed schedule for all layers. In this section, we explore a dynamic scheduler to alternate between full and recycling attention steps. Intuitively, if the query vector of a particular layer and head for the current step is similar to the query vector of the most

---

[3] https://huggingface.co/datasets/togethercomputer/RedPajama-Data-1T

Table 5: Results on continued pre-training LLaMA-3.1. The context length is 8K and we decode non-vanilla methods with $K = 2048$. We report perplexity on the last 50 tokens.

| Model | Method | Stride | Dev PPL | Test PPL | RULER Acc |
|-------|--------|--------|---------|----------|-----------|
| Base | Vanilla | - | 2.83 | 2.68 | 93 |
| + CPT | Vanilla | - | 2.83 | 2.68 | 93 |
| Base | Streaming | - | 3.20 | 3.14 | 37 |
| + CPT | Streaming | - | 3.19 | 3.14 | 37 |
| Base | Streaming++ | 50 | 3.19 | 3.13 | 37 |
| + CPT | Streaming++ | 50 | 3.18 | 3.13 | 36 |
| Base | Recycled | 50 | 3.09 | 2.96 | 83 |
| + CPT | Recycled | 50 | 3.01 | 2.87 | 84 |
| Base | Recycled | 25 | 3.03 | 2.90 | 83 |
| + CPT | Recycled | 25 | 2.97 | 2.81 | 85 |

recent full attention step, the attention pattern should be similar. Based on this, for dynamic scheduling, we only perform the full attention step when this similarity falls below a threshold.

**Approach** At every $S^{th}$ decode step, we first determine whether we *need* to perform full attention instead of always performing full attention by default. We calculate the cosine similarity between query vectors of the input token $t$ averaged across all query heads in layer $l$, with the averaged query vector of the most recent full attention step for that layer. If the similarity is higher than a threshold $s$, we decode with recycle cache, and otherwise use full attention for layer $l$. Our approach offers the flexibility of using different schedules for different layers, but uses the same schedule for all heads in the same layer. Since we perform this similarity check every $S$ steps, setting threshold $s = 1$ is equivalent to decoding with the fixed stride $S$. We perform the comparison only every $S$ steps to reduce computational overhead; we call this query comparison (QC) stride.

**Setup** We run experiments with Llama-3.1-8B on the Arxiv and Books corpus. As before, we report perplexity and decoding time measured on one A100 with batch size of 1 for the last 256 tokens of each test sequence. We run dynamic scheduler with two different query comparison strides $\{5, 10\}$ and a similarity threshold of 0.8. We compare against Recycled Attention with fixed strides 10 and 15. For RULER, we report performance on two tasks which require generating longer outputs (*multi-query* and *multi-value*). We run a dynamic scheduler with two different query comparison strides 10 and a similarity threshold of $\{0.8, 0.9\}$. We compare against Recycled Attention with fixed strides $\{10, 15\}$. For dynamic schedules, we report the effective stride across layers, i.e. the average stride at which full attention is performed.

**Dynamic stride strategy improves perplexity compared to fixed strategy when using similar decoding times.** Table 6 reports our results. We observe that using dynamic strides improves the performance-efficiency trade-off across all settings. Compared to fixed stride of 10 (row 2), dynamic stride with query comparison stride of 5 (row 3) achieves lower perplexity with a slightly faster decoding time on both domains. Similarly, employing a dynamic stride with query comparison stride of 10 (last row) achieves better or on-par performance with less decoding time compared to having a fixed stride of 15 (row 4). We observe a similar trend for RULER tasks. This better trade-off can be attributed to the larger effective stride size, i.e. less frequent full attention steps, that result from using dynamic schedules. Overall, our experiment demonstrates that dynamically deciding when to refresh the recycle cache can improve performance when using similar decode times.

## 7 RELATED WORK

**Efficient inference methods** There are multiple paths to improve decoding efficiency of long-context LMs. Prior work (Dao, 2024) achieves significant gain in inference latency by optimizing attention computations on GPUs. A line of work (Xiao et al., 2022) achieves efficiency through quantization of KV caches. We note that these are orthogonal to and can be combined with our approach. Other lines of work introduce changes to model architecture, which involves further training the model: Cai et al. (2024a) adds extra decoding heads to predict multiple subsequent tokens in parallel to further speed-up speculative decoding (Leviathan et al., 2022). Yen et al. (2024a) encodes

Table 6: Results comparing fixed stride and dynamic stride based on query similarity.

| Method | Schedule | Language Modeling | | | | | |
|---|---|---|---|---|---|---|---|
| | | Time | PPL *Arxiv* | Stride | Time | PPL *Book* | Stride |
| Vanilla | - | 7.63 | 2.22 | - | 7.63 | 7.07 | - |
| Recycled | Fixed | 7.17 | 2.36 | 10 | 7.17 | 7.49 | 10 |
| Recycled | Dynamic (QC = 5, s=0.8) | 7.07 | **2.32** | 25 | 7.07 | **7.42** | 24 |
| Recycled | Fixed | 6.88 | 2.41 | 15 | 6.94 | 7.53 | 15 |
| Recycled | Dynamic (QC = 10, s=0.8) | 6.86 | 2.36 | 32 | 6.83 | 7.54 | 31 |
| | | RULER | | | | | |
| | | Time | Acc *multi-query* | Stride | Time | Acc *multi-value* | Stride |
| Vanilla | - | 1.71 | 99 | - | 1.71 | 99 | - |
| Recycled | Fixed | 1.48 | 69 | 10 | 1.48 | 44 | 10 |
| Recycled | Dynamic (QC=10, s=0.9) | 1.32 | 69 | 15 | 1.32 | 44 | 17 |
| Recycled | Fixed | 1.43 | 62 | 15 | 1.43 | 37 | 15 |
| Recycled | Dynamic (QC=10, s=0.8) | 1.26 | 62 | 36 | 1.24 | 40 | 38 |

chunks of long context in parallel with an encoder model, which are used as input to the decoder model. We note that our method can be used as a training-free method, and show that it is possible to fine-tune the model to further improve the performance-efficiency trade-off.

**Dynamic KV cache**   Recent work (Sun et al., 2024) introduces a hierarchical speculative decoding method, which uses the model with a small KV cache constructed with attention pattern as draft model for the model with the full cache. While we share the motivation of using the attention pattern to construct a smaller KV cache, we directly leverages the dynamic cache to accelerate inference and study the performance-efficiency trade-off. Another line of recent work (Xiao et al., 2024a) proposes building a dynamic KV cache by mapping distant tokens into chunks and retrieving chunks that are similar to the current token, with the focus of extending the context size of the language model. Quest (Tang et al., 2024) uses the minimal and maximal key values to estimate import tokens for the query embedding of the current input token and load the KV cache of these tokens to decode.

**KV cache eviction**   As performing attention over the full KV cache imposes a high memory and computation burden, KV cache eviction methods have been extensively studied. Strategies include keeping only "sink" and recent tokens in the KV cache (Xiao et al., 2023); or tokens with high accumulative attention scores (Zhang et al., 2024). Building on the idea of attention-based eviction strategy, PyramidInfer(Yang et al., 2024b) retains different number of tokens per layer. Another line of work proposed query-aware eviction strategies, using the attention scores of the last few tokens in the prompt to select tokens to keep (Li et al., 2024; Cai et al., 2024b; Chen et al., 2024). Other works design eviction strategies based on attention patterns of different heads (Ge et al., 2024; Xiao et al., 2024b) or different layers (Yang et al., 2024b).

## 8 CONCLUSION

We propose recycled attention, an inference-time method which maintains a small, dynamic KV cache based on attention patterns of neighboring tokens. Our work follows a line of work leveraging the locality assumption during the attention computation. Instead of using locality to directly decide which tokens to attend to (only selecting nearby tokens), we recycle the attention pattern of nearby tokens, allowing more flexible and dynamic sparse attention patterns. We apply our method to two off-the-shelf long-context model and show that our method reduces inference wall-clock time while better preserving performance compared to prior methods which keep a KV cache of recent tokens. Finally, we show that continued pre-training the model with recycled attention and employing a dynamic stride can further improve the performance-efficiency trade-off.

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

Table 7: Results comparing different methods to aggregate attention scores for GQA models. We evaluate perplexity on sequences of length 16K for Llama-3.1-8B, where 4 query heads share the same KV head.

| Method | K | Stride | Agg | Arxiv PPL | Book PPL |
|---|---|---|---|---|---|
| Vanilla | - | - | - | 2.22 | 7.07 |
| StreamingLLM | 2048 | - | - | 2.62 | 7.94 |
| StreamingLLM++ | 2048 | 10 | - | 2.57 | 7.85 |
| Retrieval | 2048 | 10 | First | 2.43 | 7.62 |
| Retrieval | 2048 | 10 | Mean | 2.39 | 7.51 |
| Retrieval | 2048 | 10 | Max | **2.36** | **7.49** |

Zhenyu Zhang, Ying Sheng, Tianyi Zhou, Tianlong Chen, Lianmin Zheng, Ruisi Cai, Zhao Song, Yuandong Tian, Christopher Ré, Clark Barrett, et al. H2o: Heavy-hitter oracle for efficient generative inference of large language models. *Advances in Neural Information Processing Systems*, 36, 2024.

Yanli Zhao, Andrew Gu, Rohan Varma, Liangchen Luo, Chien chin Huang, Min Xu, Less Wright, Hamid Shojanazeri, Myle Ott, Sam Shleifer, Alban Desmaison, Can Balioglu, Bernard Nguyen, Geeta Chauhan, Yuchen Hao, and Shen Li. Pytorch fsdp: Experiences on scaling fully sharded data parallel. *Proc. VLDB Endow.*, 16:3848–3860, 2023. URL https://api.semanticscholar.org/CorpusID:258297871.

Ming Zhong, Da Yin, Tao Yu, Ahmad Zaidi, Mutethia Mutuma, Rahul Jha, Ahmed Hassan Awadallah, Asli Celikyilmaz, Yang Liu, Xipeng Qiu, and Dragomir Radev. QMSum: A new benchmark for query-based multi-domain meeting summarization. In Kristina Toutanova, Anna Rumshisky, Luke Zettlemoyer, Dilek Hakkani-Tur, Iz Beltagy, Steven Bethard, Ryan Cotterell, Tanmoy Chakraborty, and Yichao Zhou (eds.), *Proceedings of the 2021 Conference of the North American Chapter of the Association for Computational Linguistics: Human Language Technologies*, pp. 5905–5921, Online, June 2021. Association for Computational Linguistics. doi: 10.18653/v1/ 2021.naacl-main.472. URL https://aclanthology.org/2021.naacl-main.472.

# A APPENDIX

## A.1 ABLATING K AND S

Our method depends on two hyperparameters, the size of the recycle cache $K$ and the stride $S$ which governs how often we perform full attention and update the recycle KV cache. Here, we analyze the impact of varying these two values for Llama-3.1. We experiment with 100 ArXiv sequences with $L = 16,354$ and 14 RULER tasks with $L = 32,768$. Results are reported in Table 8 and Figure 4. We see that Recycled Attention outperforms baselines with similar inference time budget for both tasks. For example, Recycled Attention with $K = 2048, S = 16$ achieves better perplexity than StreamingLLM with $K = 4096$. In fact, Recycled Attention with $K = 4096$ achieves better accuracy than StreamingLLM with a larger $K = 8192$ for RULER. Overall, we find that increasing $K$ is more effective than decreasing the stride $S$. While decreasing stride $S$ generally benefits Recycled Attention, it has negligible effect on StreamingLLM++. This shows that the improvement does not merely come from performing full attention, but also from refreshing the recycle cache.

We further report the detailed breakdown of each RULER tasks in Table 9. We see that decreasing the stride benefits certain tasks, such as the multi-key version of NIAH and common word extraction.

## A.2 LONGBENCH EXPERIMENTS

We evaluate our method on eight long-context datasets from LongBench(Bai et al., 2023), covering multiple tasks: (1) **Single-document QA**: NarrativeQA(Kočiský et al., 2018); (2) **Multi-document QA**: Musique(Trivedi et al., 2022) and HotpotQA(Yang et al., 2018); (3) **Summarization**: GovReport(Huang et al., 2021) and QMSUM(Zhong et al., 2021); (4) **Few-shot learning**: TriviaQA(Joshi

Table 8: Updated: Recycled attention and baseline performances when varying $K$ and $S$ on RULER tasks with 32K context length (left) and Arxiv with 16K context length (right). We report results for LLama-3.1-8B with decoding time measured on a single A100 machine.

| Method | K | S | PPL | Time | K | S | Accuracy | Time |
|---|---|---|---|---|---|---|---|---|
| | | | *ArXiv Perplexity (16K)* | | | | *RULER performance (32K)* | |
| Vanilla | - | - | 2.22 | 7.39 | - | - | 90 | 1.71 |
| Streaming | 2048 | - | 2.62 | 6.64 | 4096 | - | 22 | 1.23 |
| H$_2$O | 2048 | - | 2.48 | 10.77 | 4096 | - | 21 | 2.15 |
| SnapKV | 2048 | - | 2.48 | 7.77 | 4096 | - | 72 | 1.64 |
| Streaming++ | 2048 | 32 | 2.61 | 6.61 | 4096 | 50 | 22 | 1.25 |
| Recycled | 2048 | 32 | 2.48 | 6.72 | 4096 | 50 | 63 | 1.27 |
| Recycled (k=7) | 2048 | 32 | **2.44** | 6.71 | 4096 | 50 | **79** | 1.26 |
| Streaming++ | 2048 | 16 | 2.59 | 6.77 | 4096 | 10 | 22 | 1.4 |
| Recycled | 2048 | 16 | 2.40 | 6.90 | 4096 | 10 | 65 | 1.48 |
| Recycled (k=7) | 2048 | 16 | **2.36** | 6.96 | 4096 | 10 | **82** | 1.48 |
| Streaming | 4096 | - | 2.44 | 6.94 | 8192 | - | 26 | 1.46 |
| Streaming++ | 4096 | 32 | 2.43 | 7.05 | 8192 | 50 | 26 | 1.47 |
| Recycled | 4096 | 32 | 2.33 | 7.12 | 8192 | 50 | 70 | 1.48 |

Table 9: NEW: Per-task performance of Llama-3.1-8B on RULER subtasks. For non-vanilla methods, we set the $K = 4096$ and ablate of $S$.

| Method | stride | time | niah_single | multi_key | multi_query | multi_value | fwe | vt | cwe | qa |
|---|---|---|---|---|---|---|---|---|---|---|
| | | | *Context size: 32K* | | | | | | | |
| Vanilla | - | 1.71 | 100 | 98 | 99 | 99 | 93 | 99 | 65 | 61 |
| H$_2$O | - | 2.15 | 7 | 7 | 6 | 6 | 78 | 38 | 39 | 34 |
| Streaming | - | 1.23 | 8 | 13 | 13 | 13 | 93 | 12 | 4 | 42 |
| StreamingLLM++ | 50 | 1.25 | 8 | 13 | 13 | 13 | 93 | 10 | 5 | 43 |
| StreamingLLM++ | 10 | 1.40 | 8 | 13 | 13 | 13 | 93 | 12 | 5 | 43 |
| SnapKV | - | 1.64 | 77 | 68 | 99 | 98 | 83 | 99 | 56 | 61 |
| Recycled | 50 | 1.25 | 98 | 35 | 59 | 37 | 90 | 99 | 20 | 59 |
| Recycled | 10 | 1.48 | 99 | 37 | 69 | 44 | 90 | 99 | 22 | 59 |
| Recycled (kernel=7) | 50 | 1.25 | 99 | 60 | 98 | 99 | 83 | 99 | 44 | 63 |
| Recycled (kernel=7) | 10 | 1.48 | 100 | 73 | 98 | 99 | 84 | 99 | 45 | 63 |

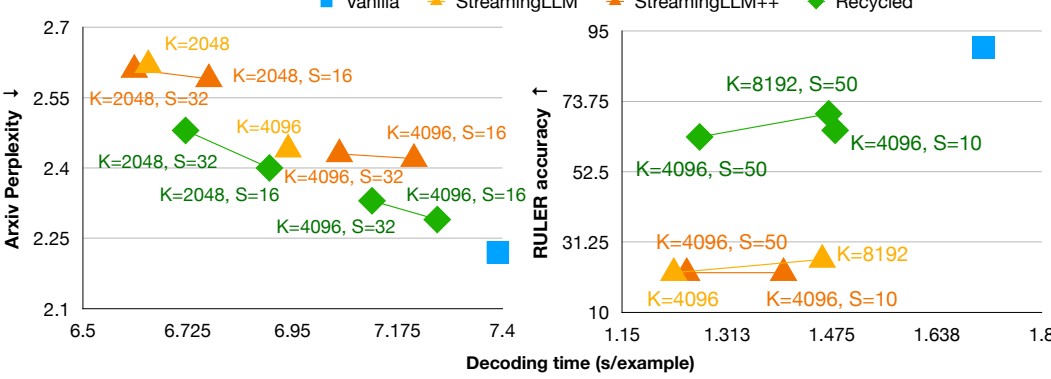

Figure 4: New: Recycled Attention and baseline performances when varying $K$ and $S$ on Arxiv with 16K context length (left) and RULER tasks with 32K context length (right) for Llama-3.1-8B. Recycled Attention achieves better performance than baselines (StreamingLLM, StreamingLLM++) with the same or less decoding time for both task.

Table 10: NEW:Performance on eight datasets with average input length greater than 5,000 from LongBench. benchmark for LLaMA-3.1-8B and Qwen-2-7B. For tasks with context length greater than 10K, we set $K = 2048$. Otherwise, we set $K = 1024$. We highlight the method with decoding time longer than vanilla in red.

| Method | stride | NarrativeQA F1↑ | t(s)↓ | Musique F1↑ | t(s)↓ | HotpotQA F1↑ | t(s)↓ | TriviaQA F1↑ | t(s)↓ | TREC Acc↑ | t(s)↓ | SAMSum R-L↑ | t(s)↓ | QMSUM R-L↑ | t(s)↓ | GovReport R-L↑ | t(s)↓ |
|---|---|---|---|---|---|---|---|---|---|---|---|---|---|---|---|---|---|
| | | | | | | | | *LLama-3.1* | | | | | | | | | |
| Vanilla | - | 40 | 2.93 | 31 | 1.38 | 46 | 0.8 | 90 | 0.83 | 70 | 1.55 | 47.70 | 3.27 | 25.17 | 14.07 | 29.29 | 13.42 |
| Streaming | - | 88 | 2.47 | 25 | 1.2 | 88 | 0.76 | 36 | 0.7 | 58 | 1.54 | 46.12 | 3.12 | 20.62 | 12.64 | 18.28 | 12.33 |
| Streaming++ | 10 | - | - | - | - | - | - | - | - | - | - | - | - | 21.18 | 13.28 | 18.88 | 13.01 |
| Streaming++ | 15 | - | - | - | - | - | - | - | - | - | - | - | - | 20.98 | 12.60 | 18.34 | 12.83 |
| H2O | - | 29 | 4.37 | 29 | 1.62 | 42 | 1.35 | 89 | 1.35 | 64 | 1.70 | 45.73 | 5.53 | 22.68 | 21.46 | 18.44 | 20.78 |
| SnapKV | - | **40** | 3.21 | **32** | 1.47 | **50** | 0.86 | **90** | 0.86 | **69** | 1.70 | 46.67 | 3.73 | 22.98 | 15.16 | 18.36 | 15.06 |
| Recycled | - | 39 | 2.56 | **32** | 1.23 | 48 | 0.7 | **90** | 0.76 | 68 | 1.54 | **47.27** | 3.15 | 23.15 | 13.22 | 19.97 | 13.00 |
| Recycled | 10 | - | - | - | - | - | - | - | - | - | - | - | - | 22.73 | 13.85 | **27.15** | 13.36 |
| Recycled | 15 | - | - | - | - | - | - | - | - | - | - | - | - | **23.24** | 12.76 | 25.96 | 13.03 |
| | | | | | | | | *QWEN-2-7B* | | | | | | | | | |
| Vanilla | - | 15 | 9.33 | 41 | 1.71 | 56 | 1.00 | **83** | 0.96 | 73 | 1.71 | 46.67 | 3.67 | 25.62 | 16.74 | 35.55 | 15.77 |
| Streaming | - | 9 | 2.30 | 33 | 1.12 | 47 | 0.71 | **83** | 0.71 | 54 | 1.59 | **46.62** | 2.92 | 21.64 | 11.84 | 20.48 | 11.06 |
| Streaming++ | 10 | - | - | - | - | - | - | - | - | - | - | - | - | 21.65 | 12.59 | 21.09 | 11.82 |
| Streaming++ | 15 | - | - | - | - | - | - | - | - | - | - | - | - | 21.65 | 12.40 | 21.39 | 11.63 |
| H2O | - | 11 | 4.03 | 28 | 1.98 | 49 | 1.16 | 87 | 1.16 | 66 | 2.45 | 45.30 | 5.21 | 23.68 | 19.13 | 22.83 | 18.75 |
| SnapKV | - | **13** | 2.86 | **41** | 1.33 | **55** | 0.85 | **83** | 0.85 | 67 | 1.67 | 46.58 | 3.36 | 25.24 | 13.37 | 26.16 | 13.61 |
| Recycled | - | **13** | 2.28 | 40 | 1.15 | **55** | 0.73 | **83** | 0.72 | **70** | 1.47 | 45.36 | 2.98 | 24.90 | 11.94 | 26.70 | 11.82 |
| Recycled | 10 | - | - | - | - | - | - | - | - | - | - | - | - | **25.44** | 12.61 | **31.01** | 12.49 |
| Recycled | 15 | - | - | - | - | - | - | - | - | - | - | - | - | 25.21 | 12.45 | 30.03 | 12.25 |

et al., 2017), TREC(Li & Roth, 2002). and SAMSum(Gliwa et al., 2019). We additionally report results for datasets with input length less than 5K in Table 11, where we observe less inference speed-up.

**Setting** We report performances for Recycled Attention and baseline approaches, as discussed in Section 3. For dataset with average context length greater than 10K (NarrativeQA, Musique, QMSum), we set $K = 2048$ for all methods. For others, we set $K = 1024$. We report SnapKV with recent window size of 32 and kernel size of 7; and Recycled Attention with kernel size of 7. For each task, we set the maximum tokens to generate following prior work (Li et al., 2024). As refreshing the recycle cache benefits longer generation (e.g. summarization), we set the stride for Recycled Attention to generation length (no refresh during generation) and additionally report a stride of {10, 15} for QMSum and GovReport. We also report a stride of {10, 15} for the StreamingLLM++ baseline. For datasets with less than 5K context, we report the result for Recycled Attention with stride at generation length, as decreasing the stride will further slow decoding down.

**Results** Experiment results are reported in Table 10. We find that Recycled Attention performs comparably or better compared to SnapKV (the best performing baseline), with faster decoding speed. SnapKV is slower than vanilla decoding in most of the cases for LLaMA-3.1-8B, as it is primarily designed for memory efficiency, instead of decoding speed.

We see that Recycled Attention outperforms SnapKV and baseline methods for the two tasks that require long generation (QMSUM and GovReport), especially when we decrease the strides. This demonstrates that maintaining the full KV cache enables the model to flexibly leverage comprehensive information in the context, which might have been evicted by methods such as SnapKV.

Results for LongBench datasets with less than 5K input length is reported in Table 11. Overall, we see a similar trend but with less efficiency gains.

### A.3 CONTINUED PRE-TRAINING IMPLEMENTATION DETAILS

**Implementation details** We train Llama-3.1 for one epoch with a global batch size of 64 and a learning rate of 5e-6. We use 20 warm-up steps and a linear schedule with 0 weight decay. We use the AdamW Optimizer. We use Fully Sharded Data Parallel (Zhao et al., 2023) and 8-bit optimizer (Dettmers et al., 2021) to improve training efficiency. Training is done on 4 H100 80 GB GPUs.

Table 11: NEW:Performance on five datasets with average input length smaller than 5,000 from LongBench for LLaMA-3.1-8B and QWEN-2-7B. We set $K = 1024$. We highlight the method with decoding time longer than vanilla in red.

| Method | stride | Qasper | | MFQA | | 2WMQA | | MultiNews | | RepoBench | |
|---|---|---|---|---|---|---|---|---|---|---|---|
| | | F1 ↑ | t(s) ↓ | F1 ↑ | t(s) ↓ | F1 ↑ | t(s) ↓ | R-L ↑ | t(s) ↓ | Sim ↑ | t(s) ↓ |
| *LLama-3.1* | | | | | | | | | | | |
| Vanilla | - | 26 | 1.52 | 46 | 1.55 | 27 | 1.57 | 26.12 | 12.13 | 49.73 | 1.70 |
| Streaming | - | 16 | 1.53 | 32 | 1.53 | 20 | 1.53 | 23.64 | 12.38 | 44.35 | 1.56 |
| $H_2O$ | - | 25 | 2.76 | 31 | 2.81 | **28** | 2.77 | **25.54** | 16.39 | **52.22** | 2.09 |
| SnapKV | - | **27** | 1.86 | **45** | 1.88 | 24 | 1.85 | 24.95 | 15.04 | 48.31 | 1.91 |
| Recycled | - | **27** | 1.55 | 42 | 1.54 | 27 | 1.55 | 25.07 | 12.42 | 47.25 | 1.58 |
| *QWEN-2-7B* | | | | | | | | | | | |
| Vanilla | - | 39 | 1.63 | 51 | 1.69 | 56 | 0.85 | 24.83 | 12.56 | 50.79 | 1.90 |
| Streaming | - | 30 | 1.45 | 26 | 1.46 | 48 | 0.72 | 22.49 | 11.86 | 45.52 | 1.46 |
| $H_2O$ | - | 33 | 2.51 | 34 | 2.53 | 46 | 1.24 | **25.11** | 15.21 | 51.14 | 2.54 |
| SnapKV | - | 35 | 1.66 | **49** | 1.67 | 54 | 0.83 | 24.43 | 13.43 | **51.58** | 1.70 |
| Recycled | - | **37** | 1.48 | 48 | 1.48 | **55** | 0.73 | 24.53 | 11.96 | 45.12 | 1.48 |

## A.4 ATTENTION PATTERN ANALYSIS

We analyze the **recovery rate** of recycled attention and StreamingLLM for LLaMA-3.1 and QWEN-2 (similar to the setting in Section 2). Figure 6 shows the aggregated attention recovery rate. We observe a consistent trend across the two domains. While for both models recycled attention recovers more attention mass than StreamingLLM, the gap between the two methods is much smaller for QWEN-2.

## A.5 DYNAMIC STRIDE ANALYSIS

We reported an aggregated effective stride in Table 6. We further investigate the effective stride patterns across different layers, shown in Figure 7. We can see that our method enables setting a different stride at different layers, with the earlier layer having a larger stride.

## A.6 RULER CONFIGURATION

We follow the suite of evaluation tasks introduced in Hsieh et al. (2024), which consists of the 13 tasks[4]. We group them based on the types:

**Single NIAH** An NIAH-styled task with one key and one value to retrieve. We include three variations of the task with different types of key, value and haystack.

**Multi-key NIAH** An NIAH-styled task with distracting keys. We include three variations of the task with different types of key, value and haystack.

**Multi-values NIAH** An NIAH-styled task with multiple values corresponding to the key.

**Multi-queries NIAH** An NIAH-styled task with multiple queries, each corresponding to a distinct key.

**Variable Tracking** A NIAH-styled task that requires tracing through multiple hops.

**Common word extraction** and **Frequent word extraction** require extracting the words based on the pattern in a list of words.

---

[4]https://github.com/hsiehjackson/RULER

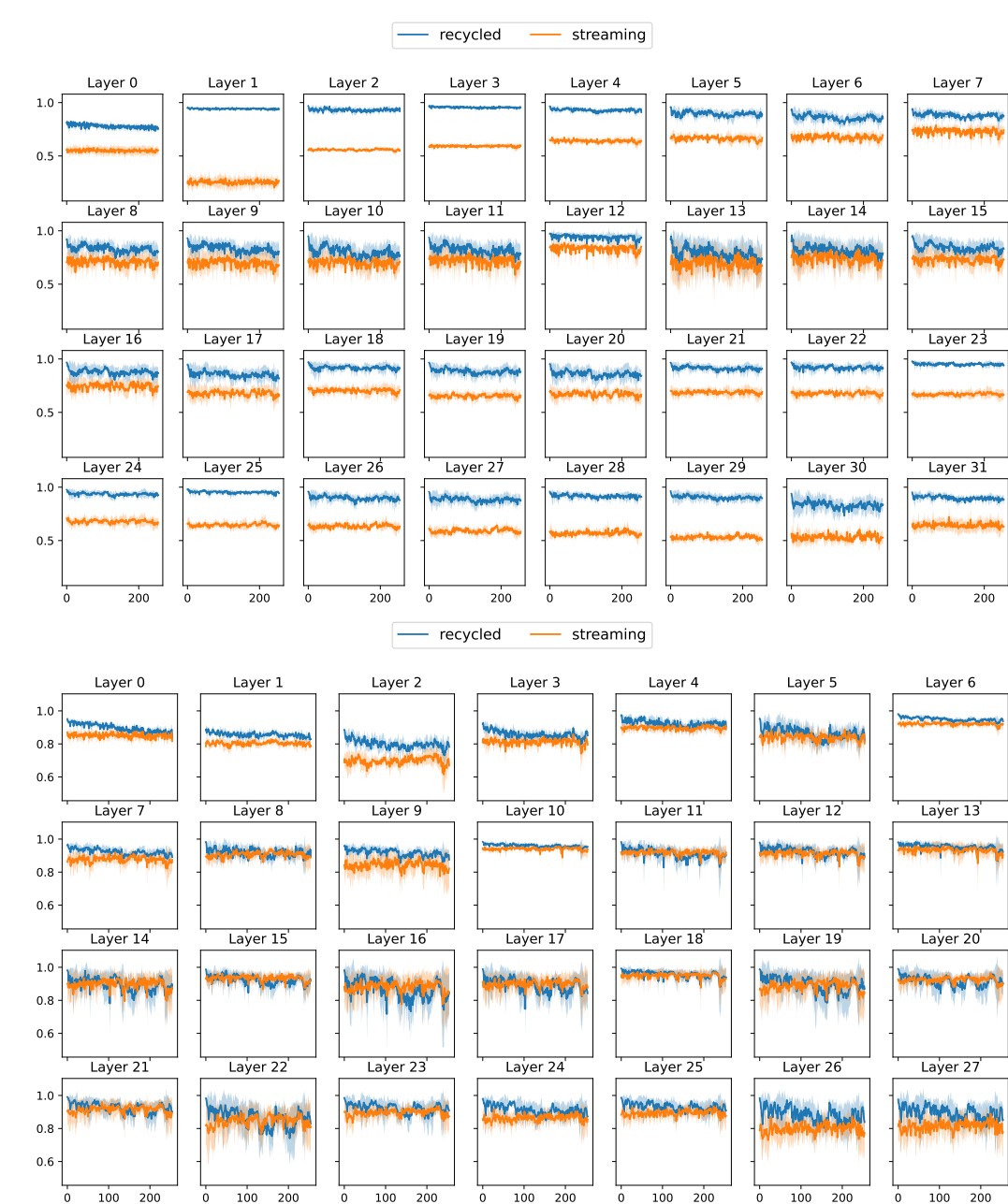

Figure 5: Recovery rate of StreamingLLM and recycled attention on 5 samples from the Arxiv split of RedPajama (Left: LLaMA-3.1, Right: Qwen-2). We calculate recovery rate with a prefill length of 8K, $K$ of 1024 and $S = 256$.

**Question Answering**   A task that requires answering a question given a set of documents. We include two variations of the tasks, corresponding to two question answering datasets.

We refer the readers to Hsieh et al. (2024) for detailed description and examples of each task.

### A.7   NEW: CHAIN OF KEY EXPERIMENTS

As most of the downstream tasks do not require long-form generation, we design a synthetic task where model needs to generate a long sequence leveraging various information in the context. We name it "chain-of-key" and provide the definition below.

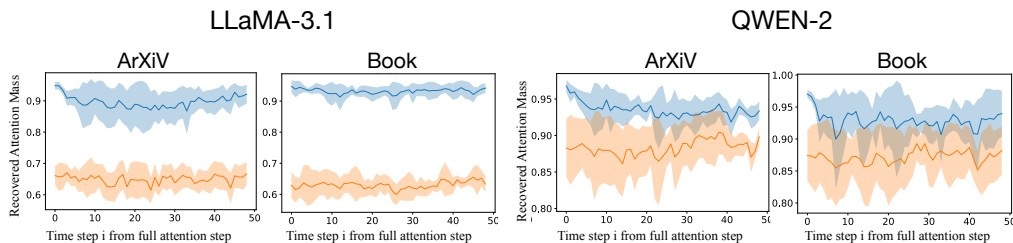

Figure 6: Recovery rate of StreamingLLM and recycled attention on 5 samples each from the Arxiv and Book split of RedPajama (Left: LLaMA-3.1, Right: Qwen-2). We calculate recovery rate with a prefill length of 8K, $K$ of 1024 and $S = 50$.

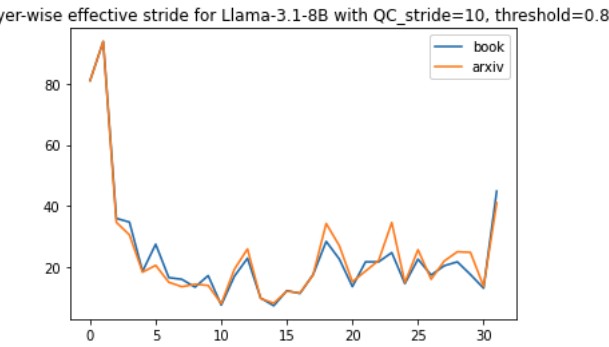

Figure 7: Layer-wise effective stride for LLaMA-3.1-8B with query similarity dynamic strides.

**Task set-up** The model is provided with context which consists of names of keys, each of which contains $W$ number of words, for instance: `apricot-waggish` where $W = 2$. The model is tasked to generate a sequence which consists a list of $T$ keys from the context, such that the first word of the next key is the last word of the current key. For example: `waggish-fishery, fishery-mosquito, mosquito-perfume, perfume-panda, panda-juice, juice-willow` for $T = 6$. We provide an example input in Table 13.

**Evaluation** We evaluate correctness of the generated output by the length of a valid chain, divided by $T$. A valid chain needs to satisfy two criteria: (a) the key must be in the context and (b) the first word of the current key must be the last word of the previous key. We provide example outputs and their correctness score in Table 14.

**Experiment setting** We report performance for all baseline methods in Section 3 as well as decoding time for all methods. Our initial experiment showed that LLaMA-3.1-8B is unable to perform this task. We thus conduct experiment with LLaMA-3.1-70B model. Decoding time is measured on 4 A100 with batch size of 1 for all methods. For Recycled Attention and StreamingLLM++, we experiment with $S = \{5, 10, 15, 20\}$.

**Results** Results are reported in Table 12. We find that Recycled Attention consistently outperformed baselines that evicts token from KV cache (SnapKV, StreamingLLM) as well as the StreamingLLM++ method, which perform full attention occasionally. SnapKV achieves an accuracy of 0.11, meaning that it is only able to generate a valid key for the first step. We also find that decreasing stride consistently improves performance for Recycled Attention and StreamingLLM++, with Recycled Attention outperforming StreamingLLM++ at every stride. Yet, having a stride that is too small (e.g. 5) might increase decoding time due to computational overhead.

Table 12: NEW:Results for the chain-of-key task of LLaMA-3.1-70B. We highlight the method with decoding time longer than vanilla in red.

| Method | K | Stride | Score↑ | Time↓ |
|---|---|---|---|---|
| Vanilla | - | - | 0.53 | 13.78 |
| StreamingLLM | 4096 | - | 0.03 | 12.23 |
| SnapKV | 4096 | - | 0.11 | 14.43 |
| StreamingLLM++ | 4096 | 20 | 0.03 | 12.41 |
| Recycled | 4096 | 20 | 0.14 | 12.88 |
| StreamingLLM++ | 4096 | 15 | 0.03 | 12.47 |
| Recycled | 4096 | 15 | 0.17 | 13.21 |
| StreamingLLM++ | 4096 | 10 | 0.04 | 12.61 |
| Recycled | 4096 | 10 | **0.19** | 13.69 |
| StreamingLLM++ | 4096 | 5 | 0.06 | 12.82 |
| Recycled | 4096 | 5 | **0.38** | 15.20 |

Table 13: Example input for the chain-of-key task where $W = 2$ and $T = 10$.

**Input**

"You are given many keys composed of a few words. Your task is to generate a chain of 10 keys such that the first word of the current key is the last word of the previous key. Separate the keys with comma. Example: waggish-fishery, fishery-mosquito, mosquito-perfume, perfume-panda, panda-juice, juice-willow, willow-bronco, bronco-creditor, creditor-bathhouse, bathhouse-woman. You must generate keys that are in the context. DO NOT REPEAT THE EXAMPLE.
Context:Name of key: toga-roommate
Name of key: appetiser-cenario
Name of key: normalization-tacit
Name of key: intensity-ping
Name of key: innate-cummerbund
Name of key: tentacle-lining [...omitted...]
Name of key: breath-yielding
Name of key: schema-festive
You are given many keys composed of a few words. Your task is to generate a chain of 10 keys such that the first word of the current key is the last word of the previous key. Separate the keys with comma.You must generate keys that are in the context. Chain of ten keys:"

Table 14: Example output for the chain-of-key task where $W = 2$ and $T = 10$ and their score. Keys that are not in the context are highlighted in red.

| Output | Score |
|---|---|
| impossible-crawdad, crawdad-vehicle, vehicle-uncertainty, uncertainty-welfare, welfare-outrigger, outrigger-historical, historical-gator, gator-hugger, hugger-debris, debris-precious | 1 (fully correct) |
| annoying-pentagon, pentagon-fit, fit-waggish, waggish-fishery, fishery-mosquito, mosquito-perfume, perfume-panda, panda-juice, juice-willow, willow-bronco | 0.2 (correct up to the second key) |
| impossible-crawdad, crawdad-vehicle, vehicle-uncertainty, welfare-outrigger, outrigger-historical, historical-gator, gator-hugger, hugger-debris, debris-precious, uncertainty-welfare | 0.3 (correct up to the third key) |

### A.8 Limitations and Future Work

**Proposed method** While we focus on accelerating inference speed, our method does not reduce memory requirement for using long-context LLMs, which can be a bottleneck for certain use cases. Our method is focused on a setting where we generate long output given a long input. When the output length is very small, the efficiency gain will be minimal. In this study, we focus on employing a fixed stride across all layers and explore dynamic scheduling based on query-similarity. Setting a custom stride per layer, or exploring other methods for deciding when to recycle the cache could be future avenue to improve performance.

**Experimental Settings** We have conducted experiment with two open-sourced long-context models and two evaluation tasks setting. We did not test out more language models and other long-context benchmarks (An et al., 2023; Karpinska et al., 2024) given our limited compute resources. Finally, our method is not limited to the language domain. Future work can explore applying recycled attention to other modalities, for instance, vision transformers.

