# OpenReview forum: "Recycled Attention: Efficient inference for long-context language models"
_ICLR.cc/2025/Conference — Submitted to ICLR 2025_

### Official Review · Reviewer_iC8W · 2024-10-20

**Soundness:** 3
**Presentation:** 3
**Contribution:** 2
**Rating:** 6
**Confidence:** 4

**Summary:**

This paper focuses on accelerating generation speed when dealing with long-context inputs using Large Language Models. Though prior methods have achieved inference speedup by different KV cache eviction policies, the eliminated tokens cannot be recovered in future generation process, thereby leading to performance decline on tasks requiring aggregating long-context information. To address this issue, the authors propose Recycled Attention, which keeps full kv cache in memory and realizes speedup by alternating between a sparse full-attention mode and a consecutive recycled attention mode. The latter only performs attention over a small subset of kv pairs identified using the attention weights produced by the full attention mode. Experiments are conducted on language modeling(Arxiv, Book, and PG19) and synthetic long-context tasks(RULER). Results demonstrate that Recycled Attention deliver higher task accuracy compared to baselines while achieving similar speedup gain.

**Strengths:**

1. The idea of utilizing attention pattern similarity between adjacent tokens to speed up inference makes intuitive sense.
2. The attention mass analysis and empirical results on downstream tasks demonstrate the effectiveness of Recylced Attention.
3. The experiment section is comprehensive, which includes two LLMs, two types of downstream tasks and different long-context configurations.

**Weaknesses:**

1. Details about the methods: Are there any discussion on the choice of using the first query head's results for each kv heads in a group? Why not the average of all query heads?
2. Benchmarks: The paper perform experiments on RULER, which is mostly a synthetic long-context benchmark and the reported results exhibit a notable margin compared to vanilla full attention. It would be more convincing to incorporate more realistic long-context benchmarks, e.e.,  LongBench.
3. Baselines: For KV cache eviction baselines, the mainly compared methods in this paper are H2O and StreamLLM, which are both query-agnostic KV cache eviction methods. The experiment lacks comparisons with more accurate query-aware KV cache eviction methods such as SnapKV, NACL, PyramidInfer, and etc, for further validation.
4. Experimental setting: The paper only focus on the decoding phase of LLM inference on short answer long-context tasks. From my understanding, a large fraction of major latency comes from the prefilling phase(Time-To-First-Token). The authors should include latency for prefilling stage to justify that the decoding speedup worth the sacrificed task accuracy. Moreover, for S=50 on RULER, it is equivalent  to prefilling-time eviction methods(such as SnapKV, NACL), of which the comparison is absent in the paper.

Reference:

[1] SnapKV: LLM Knows What You are Looking for Before Generation.

[2] NACL: A General and Effective KV Cache Eviction Framework for LLMs at Inference Time.

[3] PyramidInfer: Pyramid KV Cache Compression for High-throughput LLM Inference.

**Questions:**

See weakness above.

---

> ### Author Response · Authors · 2024-11-21
> **Author response**
>
> Thank you for your review, we are glad to see the reviewer found our work to be intuitive with comprehensive experiment settings. Please see our response below:
>
> **[W1] Attention aggregation method for GQA**
>
> We thank the reviewer for the suggestions on looking into different methods for aggregating attention scores in the same query group. We experimented with different methods to aggregate attention scores for GQA models and included a discussion in the updated manuscript (Section 3). Please see our general response.
>
> **[W2] More benchmark evaluation**
>
> We thank the reviewers for suggestions on including experiment results on LongBench. We have added results in Table 9 in Section A.2 of the updated manuscript.
>
> **[W3] More baselines**
>
> We thank the reviewers for suggesting alternative baselines of query-aware KV cache eviction strategies, which are indeed relevant to our method. We have included SnapKV as a representative approach in Table 2,3,4,9 in the updated manuscript. Please refer to our general response for detailed discussion of baselines.
>
> **[W4] Latency of pre-filling stage**
>
> The reviewer has a good point that our paper focuses on the decoding stage and does not optimize for the pre-filing stage, as we mentioned in our submission (in e.g. Section 2.1). We’d argue that this is a valid motivation, in line with previous work (e.g. SnapKV[1]), and is orthogonal to pre-filling stage optimization.

---

> > ### Author Response · Authors · 2024-12-02
> > **Follow-up on our previous response**
> >
> > Thanks for your valuable suggestions which help us improve our manuscript! We summarize our updates regarding your concerns:
> > * Regarding details of the method (aggregation for GQA models): We experimented with different methods to aggregate attention scores for GQA models and included a discussion in the updated manuscript (Section 3 and Table 7). P
> >
> > * Regarding more benchmarks: we have included 13 new datasets from LongBench (Table 10 and 11 in Section A.2) and demonstrate that our method performs on-par / better compared to baselines, especially for tasks which require longer generation (QMSum and GovReport). We have also added a synthetic task to further demonstrate scenarios where our method outperforms eviction-based method (Section A.7).
> >
> > * Regarding more baselines: we have added a new baseline (SnapKV) to all our experiments, which we believe is representative of query-aware eviction based methods.  We have also added discussion for the baselines suggested by the reviewer, and will add them as a baseline in our updated manuscript.
> >
> > * Regarding experiment settings: Please refer to our previous response for discussion of pre-filling and decoding time speed-up. For the RULER task, aside from S=50, we have also reported performance of S=10 in Table 9, which boosts the performance compared to S=50 owing to more frequently refreshing the recycled cache.
> >
> > As the discussion period approaches the end, we want to check in again and see if there are additional concerns we can address for you to consider raising the score? Thanks!

---

> > > ### Comment · Reviewer_iC8W · 2024-12-02
> > >
> > > Thanks for your response. Considering the additionally included results on dynamic striding, attention weight pooling and LongBench experiments, I believe the manuscript is stronger than before.  I have raised my rating to 6.

---

### Official Review · Reviewer_NjEN · 2024-10-27

**Soundness:** 2
**Presentation:** 4
**Contribution:** 3
**Rating:** 5
**Confidence:** 5

**Summary:**

This work aims to improve the inference speed of long-context large language models. The motivation is clear: restricting certain tokens to attend only to a subset of tokens during decoding, reduces computation and thus accelerates the decoding speed. The method is verified on a popular synthetic benchmark (RULER) and several language modeling datasets to evaluate long-context large language models.

**Strengths:**

1. The motivation and paper writing are clear. Exploring inference acceleration for long-context large language models is highly meaningful. The writing in this paper is very clear, and I can easily understand the work.
2. The method is somewhat innovative. Compared to previous similar works ([1, 2, 3]), this work considers accelerating the decoding stage.

[1] Li, Yuhong, et al. "Snapkv: Llm knows what you are looking for before generation." arXiv preprint arXiv:2404.14469 (2024).

[2] Zhang, Yichi, et al. "PyramidKV: Dynamic KV Cache Compression based on Pyramidal Information Funneling." arXiv preprint arXiv:2406.02069 (2024).

[3] Jiang, Huiqiang, et al. "Minference 1.0: Accelerating pre-filling for long-context llms via dynamic sparse attention." arXiv preprint arXiv:2407.02490 (2024).

**Weaknesses:**

1. The benchmarks are limited. The authors validated the method on RULER and several datasets for language modeling evaluation. However, there are two vital problems:
 * RULER is just a synthetic benchmark; more evaluations on real-world tasks, such as the commonly used LongBench [1], are necessary. In addition, the widely used "needle in a haystack" [2] task for evaluating long-context LLMs is necessary.
 * The authors have validated the language modeling capability of their model through extensive experiments. However, recent works ([3, 4]) have pointed out that this metric (perplexity) is not an indicative measure.
2. The baselines are limited. The authors considered two important baselines: StreamingLLM and H2O. This is reasonable because both models apply attention only to tokens within a subset of the context during decoding. However, there are many other similar works that this paper does not address, such as SnapKV [5], PyramidKV [6], MInference 1.0 [7], etc.
3. The performance loss caused by this method is too severe. From Table 2, we can see that RecycledAttention shows a significant performance drop compared to the standard model, with a decrease of 33 points on Llama 3.1 and 32 points on QWEN-2. This level of performance loss is unacceptable in practical applications. When accelerating inference speed, we should prioritize maintaining the model's performance; simply speeding up while sacrificing performance is meaningless.

[1] Bai, Yushi, et al. "Longbench: A bilingual, multitask benchmark for long context understanding." arXiv preprint arXiv:2308.14508 (2023).

[2] https://github.com/gkamradt/LLMTest_NeedleInAHaystack

[3] Gao, Tianyu, et al. "How to Train Long-Context Language Models (Effectively)." arXiv preprint arXiv:2410.02660 (2024).

[4] Hu, Yutong, et al. "Can Perplexity Reflect Large Language Model's Ability in Long Text Understanding?." arXiv preprint arXiv:2405.06105 (2024).

[5] Li, Yuhong, et al. "Snapkv: Llm knows what you are looking for before generation." arXiv preprint arXiv:2404.14469 (2024).

[6] Zhang, Yichi, et al. "PyramidKV: Dynamic KV Cache Compression based on Pyramidal Information Funneling." arXiv preprint arXiv:2406.02069 (2024).

[7] Jiang, Huiqiang, et al. "Minference 1.0: Accelerating pre-filling for long-context llms via dynamic sparse attention." arXiv preprint arXiv:2407.02490 (2024).

**Questions:**

The paper is written very clearly, and I have no questions.

---

> ### Author Response · Authors · 2024-11-21
> **Author response**
>
> Thank you for your review and suggestions. Please see our response below:
>
> **[W1] The comprehensiveness of evaluation benchmark**
>
> We thank the reviewer for suggesting an alternative benchmark, including LongBench and an needle-in-a-haystack task [0]. We would like to clarify that we test on 14 tasks in RULER, which include 8 variants of needle-in-a-haystack task. The NIAH task [0] that the reviewer is referring to corresponds to the setting of niah_single_2 in RULER (mentioned in Appendix B of RULER[1]), which is reported in the paper.
>
> For LongBench, we have included the results in Table 9 of the updated PDF, please refer to the general response for more details.
>
> The reviewer also mentioned that evaluating on perplexity is “not indicative”. While solely relying on language modeling might not comprehensively measure a model's long context performance, it is still valuable to report language modeling performance. Thus, we report a combination of language modeling perplexity and downstream tasks from RULER.
>
> [0] https://github.com/gkamradt/LLMTest_NeedleInAHaystack
> [1] RULER: What’s the Real Context Size of Your Long-Context Language Models?. COLM, 2024. https://arxiv.org/pdf/2404.06654
>
> **[W2] The comprehensiveness of baseline methods**
>
> We thank the reviewer for suggestions of alternative baseline methods. We have added experiment results for SnapKV in the updated manuscript, which we believe is the most relevant baseline for our work. Please refer to our general response for discussions on the other baseline methods.
>
> **[W3] Performance loss:**
>
> The reviewer mentioned that our approach incurs performance loss compared to the vanilla method. First, we would like to point out that our method incurs smaller performance loss compared to baseline methods. Our ablation study of varying K and S (reported in Section A.1 in the updated PDF) shows that increasing these two values can reduce performance loss at the cost of less speed-up, providing a performance-efficiency trade-off. Second, applying the max pooling methods (please refer details to Section 3.3 in the updated PDF) further closes the performance gap for both end tasks and language modeling.

---

> > ### Comment · Reviewer_NjEN · 2024-11-23
> > **Further response to authors**
> >
> > * In table 9, there are only two sub-tasks of LongBench. What's the performance over other sub-tasks?
> > * PyramidKV, Minference1.0 are lack still.
> > * The performance loss problem has not been addressed well. I think a more clear table which lists all baselines and your method over a practical benchmark such as LongBench is required.

---

> > > ### Author Response · Authors · 2024-11-28
> > > **Author response**
> > >
> > > Thank you for reading our response and the further comment. Please see our response below:
> > >
> > > **Choice of LongBench tasks:**
> > >  We experimented with these two tasks as they have context length of more than 10K, which is suitable for our setting which focuses on long-context decoding speed-up.  We have added 5 other datasets with at least 5K context in Table 10, and those that have below 5K context in Table 11. This covers all the tasks in longbench, except for LCC as the average context is only 1K and the two synthetic tasks, which we already cover with RULER. Overall, our method is on-par with baseline methods in terms of performance, and outperform them for tasks requiring longer generation (i.e. QMSum and GovReport).
> > >
> > > **Regarding new baselines**:
> > > * MInference: As we discussed in the previous general response, MInference 1.0 is a method to accelerate pre-filling stage while we target optimizing the speed for decoding stage. Please refer to section 2.1 in the manuscript for discussion on these two stages.
> > >
> > > * PyramidKV: As we discussed in the previous response, we focus on comparing to SnapKV, which represents the class of query-aware eviction methods. Such method will suffer from prematurely evicting tokens needed by future generation steps, as we have demonstrated with SnapKV. Apart from that, based on the ICLR review guidelines (https://iclr.cc/Conferences/2025/ReviewerGuide) that papers are contemporaneous if they are published within the last four months, thus we regard PyramidKV as a concurrent work. We will include it in the next version of the manuscript.
> > >
> > > **Regarding performance loss**, we summarize our performance compared to baseline model:
> > > * For the language modeling task, our method performs **the best** among the baseline, except for PG19 of QWEN-2, where StreamingLLM performs the best.
> > > * For the RULER task, our method performs **the best** and is faster than the best baseline (SnapKV). Of course, we note that SnapKV requires less memory usage, and we focus on accelerating decoding speed.
> > > * For LongBench, our method performs **on-par or better** than SnapKV, again with less decoding time.

---

> > > > ### Author Response · Authors · 2024-12-02
> > > > **Follow-up on our previous response**
> > > >
> > > > Thanks for your valuable suggestions which help us improve our manuscript! We summarize our updates regarding your concerns:
> > > >
> > > > * Regarding limited benchmarks: we have included 13 new datasets from LongBench (Table 10 and 11 in Section A.2) and demonstrate that our method performs on-par / better compared to baselines, especially for tasks which require longer generation (QMSum and GovReport). We have also added a synthetic task to further demonstrate scenarios where our method outperforms eviction-based method (Section A.7).
> > > >
> > > > * Regarding more baselines: we have added a new baseline (SnapKV) to all our experiments, which we believe is representative of query-aware eviction based methods.  We have also added discussion for the baselines suggested by the reviewer, and will add them as a baseline in our updated manuscript.
> > > >
> > > > * Regarding performance loss: Please refer to our previous response which summarizes the performance comparison between  ours and baseline methods.
> > > >
> > > > As the discussion period approaches the end, we want to check in again and see if there are additional concerns we can address for you to consider raising the score? Thanks!

---

> > > > > ### Comment · Reviewer_NjEN · 2024-12-02
> > > > > **Response to Authors**
> > > > >
> > > > > Thank you for the response. I have raised my rating.

---

### Official Review · Reviewer_kJ4i · 2024-10-31

**Soundness:** 2
**Presentation:** 3
**Contribution:** 1
**Rating:** 5
**Confidence:** 5

**Summary:**

Recycle attention combines the vanilla full attention and H2O. For every generation steps, it performs one full attention step and otherwise uses H2O. The novelty is limited.

The authors did not discuss how they decide the value of $S$ for each task. As a result, the results are questionable. Further discussion is necessary during the rebuttal period.

**Strengths:**

The paper is well-written. Many tasks and baselines are tested.

**Weaknesses:**

1. The novelty is very limited. Recycle Attention is essentially a straightforward interpolation between vanilla full attention and H2O. For every $S$ generation steps, it performs one full attention step and otherwise uses H2O. Therefore, it is not surprising that Recycle Attention performs well on the NIAH task, benefiting from occasional full attention. However, as a trade-off for combining the strengths of both approaches, Recycle Attention inherits their respective drawbacks: it is not as efficient as H2O and not as effective as full attention.

While combining vanilla full attention and H2O with an adaptive $S$ could have been novel, Recycle Attention leaves this to future study.

2. The emprically set $S$ makes experiments questionable. If I missed something, please correct me, but I did not find a detailed discussion on how $S$ was set for each task. In the ablation of S (Table 5 and Line 396), you conclude that there is "a different trend for different tasks." It appears that you set a specific $S$ for each task, making Recycle have optimal performance. If this is the case, it is a test data leakage, making the experiments unreliable.

**Questions:**

Please provide a detailed discussion on how you choose S for each task; this is crucial for evaluating the soundness of your paper. Without this information, I can only give the lowest soundness score, but I am open to revising it once you provide clear guidelines for decising the value of S.

I am also open to raising the overall rating if the authors can demonstrate that their experiments are reliable and fair.

---

> ### Author Response · Authors · 2024-11-21
> **Author response**
>
> Thank you for your review. We would like to clarify some confusions.
>
> **[W1] Is Recycled attention an interpolation of H2O and full attention?**
>
> **We would like to clarify that our method is different from switching between H2O and full attention.** H2O identifies tokens (“heavy-hitters”) based on cumulative attention scores during decoding. In contrast, our method identifies a subset of tokens to keep in the recycle cache based on the attention score of a single previous token (most recent token where full attention was performed). Thus, our method does not require access to attention scores for every decoding step, unlike H2O, which requires extra steps to compute with FlashAttention. In fact, the H2O only baseline already has a higher latency than our approach. Alternating between full and H2O would inevitably make the method less efficient.
>
> The reviewer has a good point that our method provides a middle ground between full attention and KV cache eviction method (such as H2O).  **However, we would like to clarify that our gain does not merely come from occasionally performing full attention.** One evidence demonstrated by our experiment is the performance of the StreamingLLM++ baseline, which performs full attention at the same rate of our method. Table 2 shows that its performance on RULER is significantly worse than our method and close to that of StreamingLLM. Instead, our gain comes from maintaining the full KV cache and flexibly selects the subset of tokens to attend to, based on the previous token’s attention pattern.
>
> **[Comment 1]: studying adaptive S**
>
> We thank the reviewer for suggesting to experiment with an adaptive S. We report new experiment results for adaptive S based on query similarity, please refer to our general response, and Section 6 in the updated PDF for details and results.
>
> **[W2] Selection of S**
>
> **We would like to clarify that we did not tune S based on test-set performance.** Intuitively, setting a smaller stride S will be more computationally expensive (as it entails performing full attention more often) but also more effective (as the recycle cache is refreshed more often). Thus, setting a different S provides a different performance-efficiency trade-off.
>
> We report results for a fixed S which enables empirical speed-up compared to vanilla attention. We note that we ensure all baselines have the same S to ensure fair comparison. Theoretically Recycled Attention reduces attention operation and data movement, yet setting a small stride (e.g. S=2, performing full attention every other step) does not enable speed-up empirically due to compute overheads, which also applies to baseline methods. We further reported results (for both performance and efficiency) Table 8 in Section A.1 for language modeling on arxiv and two RULER tasks, varying K and S for different tasks. This indeed shows that having a smaller S can boost performance yet at the cost of less speed-up. For instance, comparing row 3 and 4 (S=32 and 16), we see that perplexity is lower with smaller stride and yet inference time is longer. Yet, the reviewer raise a good point that it will be helpful to include a comprehensive analysis of K and S on different tasks, which we include for all 14 RULER tasks below, as well as the updated PDF.
>
> **RULER (averaged across 14 tasks) for LLama-3.1-8B**
>
> |   | Method         | K    | S  | Accuracy | Time |
> |---|----------------|------|----|----------|------|
> | 1 | Vanilla        | -    | -  | 90       | 1.71 |
> | 2 | StreamingLLM   | 4096 | -  | 22       | 1.23 |
> | 3 | StreamingLLM++ | 4096 | 50 | 22       | 1.25 |
> | 4 | Recycled       | 4096 | 50 | 63       | 1.27 |
> | 5 | StreamingLLM++ | 4096 | 10 | 22       | 1.4  |
> | 6 | Recycled       | 4096 | 10 | 65       | 1.48 |
> | 7 | StreamingLLM   | 8192 | -  | 26       | 1.46 |
> | 8 | StreamingLLM++ | 8192 | 50 | 26       | 1.47 |
> | 9 | Retrieval      | 8192 | 50 | 70       | 1.48 |
>
> We see that compared to language modeling where increasing S is beneficial for performance, increasing S is not as beneficial for RULER (comparing row [4] and [6]), compared to increasing K (comparing row [4] and row [9]). We note that this suggests different tasks might have a different set of (K,S) that will achieve the best performance-efficiency trade-off, and it is possible to choose K and S based on a small validation set.

---

> > ### Comment · Reviewer_kJ4i · 2024-11-21
> > **My concern about questionable results persists**
> >
> > Thank you for the response. My concerns persists.
> >
> > > 1. **The average accuracy on RULER is not informative.**
> >
> >    Recycle attention excels at the single-NIAH task, as shown in Table 3. It outperforms baselines on the single-NIAH task by more than 20 percent but performs poorly on other tasks. Because of the single-NIAH task, the average accuracy of your approach in Table 3 is always the highest; however, when this task is removed, your method becomes comparable to SnapKV. Therefore, the table you provided in your rebuttal is not informative, as it shows the average accuracy and does not compare the results of SnapKV, which is currently your strongest baseline.
> >
> > > 2. **My concern about questionable results persists.**
> >
> >    Of course, I understand that a smaller S increases time costs, and I acknowledge that you kept the same S for your baselines.
> >
> >    ## Let me repeat the question I am concerned about: How do you determine the value of S for each task?
> >
> >    Imagine if S=1, your method, the baselines, and the vanilla model achieve the same performance. As S increases, the performance of each approach declines. If you cannot guarantee that the performance drop rate of your method is consistently lower than that of all baselines as S increases, it is unfair to report results only when an appropriate S makes your method superior to the baselines. For instance, when S=2, as you mentioned, your method does not speed up, and if the results between your method and the baselines are comparable under S=2, then your method does not demonstrate efficiency or effectiveness advantages.
> >
> >    This is why I believe the results are questionable: RULER uses S=50 for 32k context, while the language modeling task uses S=10 for 16k context. Why 10 for 16k context, rather than 25, 30, or any other value? If you argue that a shorter context does not require a larger S like 50, then I would like to ask why 50 is used for 32k context? **Please provide evidence that your method’s performance decline is consistently slower than that of the baselines as S increases.** Otherwise, the results and wording you have presented may be misleading.
> >
> > > 3. Time cost in Table 6
> >
> >    Regarding the newly added Table 6, I do not understand why the time cost decreases when you perform more full attention operations (when s < 1).

---

> > > ### Author Response · Authors · 2024-11-28
> > > **Author response**
> > >
> > > Thank you for the reply. Please see our response below:
> > >
> > > **RULER performance:** Thank you for your suggestions on adding detailed breakdown of each RULER task for the ablation study on $S$. We have reported detailed task breakdown for different S in Table 9, and added SnapKV to table 8 (overall performance) and Table 9 of the updated manuscript. We see that multi-key and cwe benefit the most for recycled attention with kernel size of 7 when $S=10$; when kernel size is 1, decreasing the stride also benefits multi-query and multi-value. We are also consistently faster than SnapKV with varying degrees of accuracy gains per task.
> > >
> > > **Choosing the stride $S$:** We view S and K as hyperparameters that we can tune. This is similar to beam size for decoding: larger beam size will be slower but more accurate, while smaller beam size will be faster and less accurate. For some tasks, a smaller beam size suffices while for others larger beam size benefits more. Having a hyper-parameter that allows us to consider tradeoff between efficiency and effectiveness is a strength of our method. In contrast, SnapKV, which constructs a smaller KV cache only once, does not allow such flexibility.
> > >
> > > **About having different stride for language modeling and RULER task**: The reviewer is correct in commenting that the default stride value we use for RULER benchmark (50) is different from the default stride value we use for language modeling task (10). How did we reach these different strides for two tasks? For RULER, we set $S$ to a reasonable value, 50, and ran experiments. As our method was outperforming all baselines that we considered (StreamingLLM, StreamingLLM++, H2O) in terms of performance, we did not further decrease $S$ (which will improve performance at the cost of efficiency). For language modeling, we did a small pilot study exploring the value of $S$ (2, 5, 10) and chose 10 as that achieves efficiency gain over the vanilla baseline, while smaller stride does not enable speed-up compared to vanilla, though performs better than $S=10$.
> > >
> > > We agree with the reviewer that selection of $S$ is important. Doing a more careful, systematic search of hyper parameter $S$ based on the development set performance can further improve performance of our approach per each end task it aims for, at the cost of computational resources.
> > >
> > > **Regarding the evidence that our method’s decline is consistently slower than baseline when increasing $S$**: At any stride $S$, our approach outperforms StreamingLLM++, the only baseline which allows the additional hyper-parameter $S$ to balance efficiency vs. effectiveness. We show this at stride 10, and 50 for RULER; and stride 16, 32 for language modeling tasks in Table 8. For the new experiments we added, we also reported multiple $S$ – {10, 15} for the summarization tasks in Table 10 and {5, 10, 15, 20} for the synthetic chain-of-key task in Table 12, showing that our method consistently outperforms baselines at each stride.
> > >
> > > **Regarding table 6**, the $s$ here refers to the similarity threshold which we use to decide whether to perform full attention again, while the effective stride is reported as the last column (“Stride”). We describe this in line 449 in Section 6. We apologize for the confusion, and will update the manuscript to make the notation clearer.

---

> ### Comment · Reviewer_kJ4i · 2024-11-28
> **My concerns about questionable results persist**
>
> Your response and the new results have raised further concerns about the tuning of hyperparameters based on the test data in your experiments.
>
> ### Q1
> > For language modeling, we did a small pilot study exploring the value of $S$ (2, 5, 10) and chose 10 as that achieves efficiency gain over the vanilla baseline, while smaller stride does not enable speed-up compared to vanilla, though performs better than $S=10$.
>
> As clarified in my previous comment, "when S=2, as you mentioned, your method does not speed up, and if the results between your method and the baselines are comparable under S=2, then your method does not demonstrate efficiency or effectiveness advantages." The key issue is whether the smaller stride values ($S=2$ and $S=5$) outperform the baseline, instead of whether smaller strides outperform larger strides; if they do not, the results could be seen as tuned to the test set, especially given that you did test with $S=2$ and $S=5$ but did not report those results.
>
> To address this concern, please demonstrate that your method outperforms the baselines when $S=2$ and $S=5$ to convince me (even though these settings have worse efficiency, which I acknowledge).
>
> By the way, I have acknowledged that a smaller $S$ improves performance but reduces efficiency, twice. Please avoid repeating this information to make our discussion more productive.
>
> ### Q2
>
> > At any stride $S$, our approach outperforms StreamingLLM++
>
> This seems overly broad now, especially considering the narrow range of strides ($S=5, 10, 15, 20$) used in your comparisons. To strengthen this claim, please show me the comparison with StreamingLLM++ at $S=2$.
> Because from Table 12, it appears that when $S < 10$, your method actually takes more time and performs worse than the vanilla baseline. Therefore, it is important to show whether your method can still outperform StreamingLLM++ at $S=2$, to justify the claim.
>
> ### Q3
>
> There is some confusion regarding the stride values ($S$) used in Table 6. Specifically, for the Dynamic method with $QC = 5$ and $s=0.8$, $S=25$ is used for the Arxiv dataset and $S=24$ for the Book dataset. In contrast, the Fixed baseline uses $S=10$ for both datasets. How were these stride values chosen for the Dynamic method in Table 6? If $S$ was tuned for the Dynamic method but not for the Fixed baseline, is this a fair comparison? Additionally, some of the stride values in Table 6 (e.g., 17, 31, 36, 38) seem too unusual and lack clear justification. Could you clarify how these stride values were selected?

---

> > ### Author Response · Authors · 2024-11-28
> > **Author response**
> >
> > Thank you for the reply. Please see our response below:
> >
> > **For Q1 and Q2**:
> >
> > For the new “chain-of-key” task, we have shown that our method outperforms StreamingLLM++ for $S=5$ in Table 12 (0.35 v.s. 0.05). We further report the results for $S=2$ below, where our method also outperforms StreamingLLM++ (0.54 v.s. 0.07). We also further report performances for $S=2$ and $S=5$ for the RULER tasks and  the language modeling task on Arxiv.
> >
> > On all tasks, our method outperforms StreamingLLM++ for $S=(2, 5)$ in terms of performance.
> >
> > **LLaMA-3.1-70B, chain-of-key (corresponding to Table 12 in the PDF)**
> > | Method         | K    | S | Accuracy | Time    |
> > |----------------|------|---|----------|---------|
> > | Vanilla        | -    | - | 0.53     | 13.78   |
> > | StreamingLLM++ | 4096 | 2 | 0.07     | 13.55   |
> > | Recycled       | 4096 | 2 | 0.54     | _19.76_ |
> > | StreamingLLM++ | 4096 | 5 | 0.06     | 12.82   |
> > | Recycled       | 4096 | 5 | 0.38     | _15.20_ |
> >
> > **LLaMA-3.1-8B RULER (context size: 32K, corresponding to Table 2 in the PDF)**
> > | Method         | K    | S | Accuracy | Time   |
> > |----------------|------|---|----------|--------|
> > | Vanilla        | -    | - | 90       | 1.71   |
> > | StreamingLLM++ | 4096 | 2 | 24       | _1.98_ |
> > | Recycled       | 4096 | 2 | 89       | _2.44_ |
> > | StreamingLLM++ | 4096 | 5 | 22       | 1.54   |
> > | Recycled       | 4096 | 5 | 87       | _1.72_ |
> >
> > **LLaMA-8B Language modelling on arxiv (context size: 16K; corresponding to Table 4 in the PDF)**
> > | Method         | K    | S | Perplexity | Time   |
> > |----------------|------|---|------------|--------|
> > | Vanilla        | -    | - | 2.22       | 7.63   |
> > | StreamingLLM++ | 2048 | 2 | 2.40       | _8.24_ |
> > | Recycled       | 2048 | 2 | 2.23       | _9.42_ |
> > | StreamingLLM++ | 2048 | 5 | 2.52       | 7.31   |
> > | Recycled       | 2048 | 5 | 2.26       | _7.70_ |
> >
> > Besides the empirical evidence, we provide reasoning for why a smaller stride improves the performance for Recycled Attention, but not for StreamingLLM++.
> >
> > Let’s consider the case for $S=2$, this means, for step ```i```, the model performs attention using full KV cache $C_{full}$, at step ```i+1```, the model performs attention using a smaller KV cache, $C_{small}$. For StreamingLLM++, $C_{small}$ contains the first 4 tokens (sink) and tokens from ```[i-K-4, i]```, and for Recycled Attention, $C_{small}$ contains the top K tokens which received the highest attention score from step ```i```.
> >
> > Now, at step ```i+1```, suppose we are decoding a needle which consists of multiple tokens from the needle-in-a-haystack task. If the needle is not in the recent ```K-4``` tokens (i.e. not in $C_{small}$), then StreamingLLM++ won’t be able to continue decoding it at ```i+1```. For Recycled Attention, as the needle receives high attention scores in step ```i```, it will be in $C_{small}$ for recycled cache.
> >
> > **For Q3**: We believe there is some misunderstanding. The $S$ reported in table 6 are **effective strides**, which are not set manually. As described in Section 6, we use the dynamic stride approach where query similarity decides when to perform the next full attention step. For dynamic stride, what we set is the QC stride (how often to perform the similarity check) and the query similarity threshold (which decides whether to perform full attention or not). We experimented with the threshold of [0.8, 0.9], which resulted in different effective strides in Table 6.  What you are referring to, e.g. effective stride of 25 for QC=5 and s=0.8, reflects how often the method ends up performing full attention (defined at line 463-465), which we measure post-hoc.

---

> > > ### Author Response · Authors · 2024-12-02
> > > **Follow-up on our previous response**
> > >
> > > Thanks for your suggestions and feedback to help us improve our paper. We summarize our updates regarding your concerns:
> > > * Regarding choice of S and ablation: we have added an ablation study for language modelling and RULER, including various values of S (2, 5, 10, 50 for RULER and 2, 5, 10, 16, 32 for language modeling). We have also reported performance for various S for the two new sets of tasks that we have added during rebuttal (LongBench and “chain-of-key”). Together, we show that by varying S, our method offers a performance-efficiency trade-off and our method outperforms the baseline method (StreamingLLM++) at all the S we experimented with. We believe that this addresses the key limitations raised in your review.
> > >
> > > * Regarding difference between our method and H2O: we have explained in the response that our method is not an interpolation between full attention and H2O, but a new paradigm which, instead of evicting tokens permanently from the KV cache (e.g. H2O, StreamingLLM), maintains the full KV cache and selects a subset of tokens to perform attention based on the attention pattern of neighboring tokens.
> > >
> > > * Regarding adaptive stride: we have updated our manuscript to include a section on dynamic stride (Section 6). Our experiment shows that dynamic stride achieves similar performance with faster decoding speed compared to fixed stride (Table 6), providing better performance-efficiency trade-off.
> > >
> > > As the discussion period approaches the end, we want to check in again and see if there are additional concerns we can address for you to consider raising the score? Thanks!

---

> > > ### Comment · Reviewer_kJ4i · 2024-12-02
> > >
> > > Thank you for the response. I have raised my rating.

---

### Official Review · Reviewer_PFMm · 2024-11-02

**Soundness:** 3
**Presentation:** 4
**Contribution:** 3
**Rating:** 5
**Confidence:** 4

**Summary:**

This paper proposes a new efficient LLM inference method called Recycled Attention. Observing that the top-k tokens based on attention scores at current step still hold a significant portion of the attention mass over the following steps, authors interleave full attention within sparse attention to more accurately select important tokens, balancing the advantages and disadvantages of full and sparse attention.

**Strengths:**

Previous efficient inference methods based on KV Cache compression typically do not reuse a token once it has been evicted. Recycled Attention addresses this issue by interleaving Full Attention.
Compared to Vanilla Attention, Recycled Attention significantly reduces inference latency.
Compared to StreamingLLM and H2O, Recycled Attention achieves substantial gains on the Ruler Benchmark with nearly comparable inference latency.

**Weaknesses:**

Recycled Attention increases the memory burden. Compared to StreamingLLM and H2O, Recycled Attention requires additional maintenance of a full KV Cache.
The benchmark is limited. Ruler uses synthetic examples for testing. Performance on real sample benchmarks, such as LongBench[1], should also be reported.
After continued pretraining, only PPL is tested. For long context text modeling, PPL is not an intuitive metric. Additional experiments on Ruler and LongBench are needed.
Stronger comparison methods are missing. For example, Quest[2], which has a similar motivation to this paper, should also be included for comparison.
[1] LongBench: A Bilingual, Multitask Benchmark for Long Context Understanding (Bai et al., ACL 2024)
[2] Quest: Query-Aware Sparsity for Efficient Long-Context LLM Inference (Tang et al., ICML 2024)

**Questions:**

I am curious about how H2O would perform in the attention mass overlap test in Section 2.2.
Additionally, most tokens in the Full KV Cache also come from the Recycle Steps. As the generation length increases, will there be an accumulation of errors compared to Vanilla Attention, resulting in an increasing performance gap between Recycled Attention and Vanilla Attention?

---

> ### Author Response · Authors · 2024-11-21
> **Author response**
>
> Thank you for your review! We are encouraged to see that the reviewer found our work to achieve better performance compared to previous method. Please see our response below:
>
> **[W1] Limited benchmark**
>
> We thank the reviewers for suggesting alternative benchmarks such as LongBench for our experiments. While it is true that RULER primarily contains synthetic tasks such as NIAH, we note that it also contains realistic tasks such as question answering. We have included LongBench results in Table 9 in the updated PDF and please refer to our general response for more discussion.
>
> **[W2] Limited evaluation of continued pretraining (CPT)**
>
> We thank the reviewer for suggestions to evaluate the CPT model on the RULER task, which we have updated in Table 6 in the updated PDF. We observe a similar improvement as the language modeling task. We note that we continued pre-trained the model on a relatively small amount of tokens due to our limited compute resource, and it is possible that further CPT can lead to more gains.
>
> **[W3] More baselines.**
>
> We thank the reviewer for suggesting QUEST as an alternative baseline. We have added another baseline (SnapKV), which is more relevant for our method and we are working on adding a comparison to QUEST. Please refer to our general response about discussion on alternative baselines.
>
> **[Q1] H2O performance on attention mass overlap.**
>
> Thank you for suggesting to add H2O’s performance on attention mass overlap in section 2.2. We have included it in figure 2 in the updated manuscript. H2O performs better than StreamingLLM, as reflected in our language modeling experiments.
>
>
> **[Q2] Error pattern as the generation length increase:**
>
> We do not anticipate there will be accumulation of error, as we recycle the attention pattern from the nearest token (at the maximum, S-1 steps away in a fixed scheduling setting). This is supported by our language modeling experiments, for which we report performance on the last 256 tokens.

---

> > ### Author Response · Authors · 2024-11-28
> > **Follow-up on previous author response**
> >
> > Dear Reviewer PFMm, we want to check in to see if our previous response has addressed your concern. We also want to provide a further comment on comparison with QUEST:
> >
> > * After our investigation, we found that their implementation without kernel is pretty slow, while the kernel implementation currently does not support GQA models (LLaMA-3.1-8B and QWEN-2-7B we considered in our experiments) yet. Hence it is difficult to make a comparison. Their method presents a different way to select a subset of the full KV cache to move and attend to, and it is possible to combine our method with QUEST (e.g. refreshing the Top-K critical KV cache pages every S step).

---

> > > ### Comment · Reviewer_PFMm · 2024-11-28
> > >
> > > Thanks for your response! I appreciate the additional benchmarks and baselines in the experimental section.
> > > I still have one question: In the attention mass overlap experiment, after incorporating H2O, it is observed that the performance of H2O is almost identical to that of Recycled Attention. This suggests that the performance improvement of Recycled Attention over the H2O method is not due to an increase in the attention mass recovery rate. Could you provide more specific examples comparing the differences in token maintenance within the KV Cache between Recycled Attention and H2O?

---

> ### Author Response · Authors · 2024-11-28
> **Author response**
>
> Thank you for your question! We conducted the attention mass overlap experiment on the arxiv dataset for Figure 2, and you are right that the performance of H2O and Recycled Attention is close. This is indeed reflected in the language modeling performance in Table 4 (note that we reported performance for S=10 in Table 4; but we did not refresh the Recycled Attention in the attention mass analysis, hence S=50).
>
> However, if we consider the needle-in-a-haystack task from RULER and conduct the attention mass analysis for generating the output, for context length of 8192 and $K=1024$, Recycled Attention recovers over 97% of attention mass while StreamingLLM and H2O recovers less than 90%, as reflected in the results of RULER experiments. This is because H2O uses cumulative attention score to decide tokens to keep and might have evicted the target value from the KV cache. Thank you for the suggestion on analyzing attention mass overlap in different scenarios and we will add this to our updated manuscript.

---

> > ### Author Response · Authors · 2024-12-02
> > **Follow-up on our previous response**
> >
> > Thanks for your valuable suggestions which help us improve our manuscript! We summarize our updates regarding your concerns:
> > * Regarding limited benchmarks: we have included 13 new datasets from LongBench (Table 10 and 11 in Section A.2) and demonstrate that our method performs on-par / better compared to baselines. We have also added a synthetic task to further demonstrate scenarios where our method outperforms eviction-based method (Section A.7).
> >
> > * Regarding more baselines: we have added a new baseline (SnapKV) to all our experiments, which we believe is representative of query-aware eviction based methods. While the reviewer is correct that our method requires more memory compared to eviction-based methods, we have demonstrated that our method performs better than eviction-based methods for longer generation (QMSum and GovReport, as well as a new synthetic task which we have added to Section A.7). We have also provided a discussion about our method and QUEST in our previous response.
> >
> > * Regarding more evaluation for CPT: we have added RULER results in Table 5, showing that CPT brings improvement to RULER, besides improvement on language modelling from the initial submission.
> >
> > * Regarding attention overlap analysis for H2O: we have added performance of H2O in Figure 2 (tested on arxiv), and the performance of Recycled Attention, StreamingLLM and H2O on the NIAH task in our previous response, which we will add in our updated manuscript.
> >
> > As the discussion period approaches the end, we want to check in again and see if there are additional concerns we can address for you to consider raising the score? Thanks!

---

### Official Review · Reviewer_h8fr · 2024-11-02

**Soundness:** 3
**Presentation:** 3
**Contribution:** 3
**Rating:** 6
**Confidence:** 4

**Summary:**

This paper tries to tackle the challenges in long context LLMs. Long context LLMs create lots of KV cache during inference, therefore requiring large memory space and high bandwidth requirements. A representative line of work to address this problem is aiming at reducing the number of KV cache stored during inference. However, the authors reported performance problems for these approaches. Therefore, they proposed recycled attention. Instead of completely getting rid of some KV caches, a full attention is performed periodically generation process and a partial attention (i.e. using fewer important KV cache) is performed for the rest of the time. Evidences show that recycled attention can effectively identify the important KV cache and achieves higher performance without sacrificing too much efficiency.

**Strengths:**

1. The authors have used recovery rate to justify the idea of recycled attention. I do think the idea itself presents a thinking of hierarchical attention subject to generation phase.
2. The presentation of this paper is clear and convincing.

**Weaknesses:**

1. It would be good to have theoretical reasoning on the effectiveness of the recycled attention idea.
2. As mentioned in the Limitation section, it would improve the paper if the experiments can go deeper into more settings, like the different stride for different layers. These questions are likely to raised after reading the paper. It is worth to have these results and will be helpful to draw more comprehensive insights. Otherwise, the contribution of the idea seems pretty limited.

**Questions:**

1. Why QWEN appears to get a similar performance for streaming LLM and recycled attention? This pattern is true for both Figure 4 and FIgure5.
2.  Why the gain is minimal when the output length is small? I feel like if the output length is small, it is possible that only the first step attention is in full form, others should be fast in terms of generating the tokens.

---

> ### Author Response · Authors · 2024-11-21
> **Author response**
>
> Thank you for your review and we are glad to see the reviewer found our work to be intuitive and convincing. Please see our response below:
>
> **[W1] Theoretical reasoning on effectiveness of Recycled Attention**
>
> We have empirical support for our method. Our key intuition for recycled attention is that neighboring tokens are likely to place attention mass on similar sets of tokens in the context. And this is justified by the recovery rate of attention mass when inference with recycled attention (fig 2), as mentioned by the reviewer.
>
> **[W2] Setting a different stride for different layers**
>
> We thank the reviewer for suggestions on exploring dynamic stride and report new experiment results for dynamic scheduling based on query similarity in section 6 of the updated PDF, please also refer to the general response. In response to the reviewer’s suggestion of setting a different stride at different layers, we have included an analysis on per-layer effective stride in Section A.6 in the appendix, finding that earlier layer having a larger stride.
>
>
> **[Q1] QWEN’s recovery rate:**
>
> As the reviewer mentioned, Figure 4 shows that recycled attention’s recovery rate is closer to (although still outperforming) StreamingLLM, compared to LLaMA-3.1-8B. We included the analysis to understand differences between the model, and hypothesized that it might be why we observed better performance for LLaMA-3.1-8B with Recycled Attention compared to QWEN-2.
>
> **[Q2] Generation setting:**
>
> What we meant by “output length is very small, the efficiency gain will be minimal (line 486)” refers to a setting where the target output length is smaller than the stride (e.g., LLM replies single word response such as “yes” or “no”). We will clarify this in the revision. In most use cases, however, LLM generates long-form responses, in which case the efficiency gain depends on the stride S at which full attention is performed, instead of the length of the tokens generated.

---

> > ### Comment · Reviewer_h8fr · 2024-11-23
> >
> > Thank you for the detailed comment and updated results on dynamic scheduling.

---

> > > ### Author Response · Authors · 2024-12-02
> > > **Follow-up on our previous response**
> > >
> > > Thanks for your valuable suggestions which help us improve our manuscript!
> > >
> > > We have updated our manuscript to include a section on dynamic stride (Section 6), which enables dynamically setting a different stride per different layer, as the reviewer suggested. Our experiment shows that dynamic stride achieves similar performance with faster decoding speed compared to fixed stride, providing better performance-efficiency trade-off. We believe this addresses the key limitation raised in your initial review.
> > >
> > > As the discussion period approaches the end, we want to check in again and see if there are additional concerns we can address for you? Thanks!

---

### Author Response · Authors · 2024-11-21
**General response 1/2**

We thank all reviewers for their reviews and helpful comments. We are delighted to see that they found our work to present an intuitive and novel idea (Reviewer h8fr
, Reviewer PFMm, Reviewer NjEN, Reviewer iC8W), with clear motivation and presentation (Reviewer kJ4i, Reviewer NjEN, Reviewer iC8W) and achieving substantial gains compared to previously proposed methods (Reviewer PFMm). Here we include new experiment results and clarifications regarding common requests from the reviewers.

We have uploaded a new manuscript with new experiment results / discussion (highlighted in blue text or yellow background) and updated results (highlighted in red).

**Summary of changes made to the manuscript**:
* New baselines added (SnapKV). See Section 3.2.
* Results added on LongBench datasets, in Appendix section A.2.
* Dynamic stride selection based on the similarity of queries of the current time step with that of the last full attention step, in Section 6.
* Discussion of the suggested baseline in the related work section (Section 7).
* Discussion of aggregation methods for GQA models, in Section 3. We have also updated the experiment results for Recycled Attention with max aggregation (instead of first in the group) in all the tables.
* RULER results for continued pre-training, see Table 5 in Section 5.

We elaborate on these below.


**More baselines**:
We thank the reviewers for suggesting other relevant baselines (Reviewer PFMm, Reviewer NjEN, Reviewer iC8W) from recent work. Here we provide a discussion on them. We have also added a paragraph in the related work section of the updated PDF and updated result table to include suggested baseline (SnapKV).

* (1) Query-aware permanent KV cache eviction method:
SnapKV[1]  (Reviewer iC8W, NjEN), NACL[2] (Reviewer iC8W)  and PyramidKV[3] (Reviewer NjEN) are query-aware KV-cache eviction methods. Among these, SnapKV is the most relevant to our method, as it uses attention scores of the last few tokens in the prompt to select tokens to keep. We focus on comparing against SnapKV and include the new experimental results in Table 2,3,4,9. Our method outperforms / performs on-par with SnapKV for both the language modeling and downstream tasks (RULER and LongBench), with a slightly faster decoding speed.

* (2) Quest [5] (suggested by Reviewer PFMm): Unlike most prior work, our method dynamically selects tokens that are likely to be relevant at the current generation step. Quest [5] is the only other method that maintains the full KV cache and dynamically constructs a smaller KV cache for attention computation. While we leverage previous tokens’ attention pattern, they use the minimal and maximal key values to estimate import tokens for the current input token. This method incorporates PageAttention and Top-K cuda filtering, making inference speed comparison a bit challenging. We are working on adding this baseline as a comparison.

* (3) MInference [6] (suggested by Reviewer iC8W). This approach accelerates the pre-filling stage while we focus on accelerating the decoding stage, so not very applicable in our setting as a baseline.

* (4) PyramidInfer [4] (suggested by Reviewer iC8W): This is a query-agnostic method and leverages accumulated attention scores to evict tokens during both the pre-filling and generation stage, similar to the H2O baselines that we have included. As demonstrated by our experiments of H2O, such query-agnostic KV cache eviction methods can prematurely evict tokens.


[1] SnapKV: LLM Knows What You are Looking for Before Generation. NeurIPS 2024.
[2] NACL: A General and Effective KV Cache Eviction Framework for LLMs at Inference Time. ACL 2024
[3] PyramidKV: Dynamic KV Cache Compression based on Pyramidal Information Funneling. Arxiv, 2024.
[4] PyramidInfer: Pyramid KV Cache Compression for High-throughput LLM Inference. ACL 2024.
[5] Quest: Query-Aware Sparsity for Efficient Long-Context LLM Inference. ICML 2024.
[6] MInference 1.0: Accelerating Pre-filling for Long-Context LLMs via Dynamic Sparse Attention. NeuRIPS 2024.

**Evaluate on more benchmark datasets**:
Reviewers (PFMm, NjEN, iC8W) asked for results on another, “more realistic”, benchmark, specifically LongBench [1] .

First, we would like to clarify that our initial evaluation dataset, RULER contains 14 tasks. Two of which are QA tasks (reported under “QA” in Table 3) that contain realistic question and answers, in addition to synthetic tasks such as NIAH.

We include results for two datasets from LongBench (NarrativeQA and Musique) that have input length longer than 10K in Table 9 of Section A.2 in the updated PDF. Overall, we found that our method performs on-par or better with the best baseline (SnapKV) with faster decoding speed, repeating the trend we see in other benchmarks.

---

> ### Author Response · Authors · 2024-11-21
> **General response 2/2**
>
> **Dynamic scheduling:**
> Multiple reviewers (Reviewer h8fr, Reviewer kJ4i) suggested dynamic scheduling with an adaptive stride S. We experiment with dynamic scheduling based on query similarity. Concretely, instead of performing full attention at a fixed stride, we decide to perform full attention for the current decoding step or not  based on the similarity of query embeddings with the query embedding of the last full attention step. Our experiments show that Recycled Attention can be further improved with dynamic scheduling: it achieves better efficiency-performance trade-off compared to static stride (what we reported in the submitted version of the paper) for both the language modeling tasks and two RULER tasks. We describe the method, experiment setting as well as experiment results in Section 6 and Table 6 in the updated PDF.
>
> **GQA aggregation methods:**
> Reviewer iC8W asked about how to aggregate attention scores for GQA models. In our submission, we used the attention score of the first query head to choose top K tokens for the entire query group. We later experimented with taking the average and the maximal score, finding that taking the max performs the best. We have added a discussion in section 3 of the updated PDF (highlighted in blue) and included an ablation study in Table 7 in the appendix. We have also updated the results table (table 2, 3, 4) for recycled attention with max attention scores, showing better performance for both RULER and language modeling evaluation.

---

> ### Comment · Reviewer_iC8W · 2024-11-21
>
> Thanks authors for their detailed response. After reading the general response, I have some further questions listed below:
>
> 1.  The authors stated that PyramidInfer is a "query-agnostic method and leverages accumulated attention scores to evict tokens during both the pre-filling and generation stage". This is indeed not true. PyramidInfer adopt a similar strategy to SnapKV: during pre-filling stage, it uses only the weighted average of attention weights of recent sequence S_r(which is equivalent to the observation window in SnapKV) to evict unimportant KV pairs. During decoding, it employ the same strategy using a sliding recent sequence window.
> 2.  Another important distinction between RecycledAttention and compared baselines(including SnapKV) is the retention of complete KV cache. This is to say, RecycledAttention still suffer from massive memory usage when applied to large LLMs and long-context tasks. According to results on RULER and LongBench, RecycledAttention is mostly on par with SnapKV in terms of accuracy, slighter faster than SnapKV in speed. I suggest the authors also report the memory footprint of each method to comprehensively reflect the efficiency gain of each approach.
> 3. Follow the previous comment, my opinion is that since RecycledAttention focus on improving decoding speed(not memory), the paper would be much stronger if the authors could demonstrate the compatibility of RecycledAttention with other memory-efficiency techniques to fully support the claim of "efficient inference".

---

> > ### Author Response · Authors · 2024-11-28
> > **Author response to Reviewer iC8W**
> >
> > Thank you for reading our response and for the further questions! Please see our reply below.
> >
> > * Thank you for your correction on PyramidInfer! We will update our manuscript. Upon further investigation, we found that PyramidInfer currently does not support FlashAttention, thus we do not include it as a baseline in the rebuttal.
> >
> > * Yes indeed, our method will be more expensive in terms of memory compared to KV cache eviction method and we provide a comparison for both memory and time complexity of different methods in Table 1. We will further clarify the distinction of different methods (some are more memory efficient while others are more compute efficient) in the manuscript.
> >
> > * Regarding performance gap between SnapKV and Recycled attention: we have added a new experiment setting for a synthetic task which requires the model to perform long-form generation leveraging various information in the context; as well as two summarization tasks that require longer generation. There, we see further benefit of keeping the full KV cache (hence more memory usage) compared to memory-efficient eviction methods.
> >
> > * Thank you for the suggestions on combining Recycled Attention with memory-efficient techniques to enable memory efficiency. Indeed, it will be interesting to explore combining Recycled Attention with approaches such as KV cache quantization, as future work. In this paper we focus on improving decoding time efficiency, and we are happy to clarify in the title (e.g. “Faster Inference” instead of “Efficient Inference”) to avoid misunderstanding.

---

### Author Response · Authors · 2024-11-28
**General Response (Further comparison with SnapKV) [1/2]**

We thank the reviewers for their reply to our previous response. We address commonly raised questions among reviewers here and provide experiment results to compare our methods against SnapKV, the newly added baseline.

A few reviewers (Reviewer kJ4i, Reviewer NjEN and Reviewer iC8W) mentioned that our method is mostly on-par with SnapKV in terms of performance on current experiments. Our performance is on-par with SnapKV for the two LongBench datasets (NarrativeQA and Musique)  we added previously, and is better than SnapKV and language modelling, all with a faster decoding speed. Here, we further clarify our differences with SnapKV, and present further evidence for scenarios where our methods are better than SnapKV in terms of performance.

One major distinction between our method as SnapKV is that we maintain the full KV cache and occasionally perform full attention to refresh the recycle cache, while SnapKV permanently evicts tokens that might be useful in later generation. Conceptually, our method will outperform SnapKV on task settings where LLM has to leverage different tokens in the context based on the tokens that it has generated. This will apply to many real-world scenarios, such as (1) LLM is tasked with generating longer text or (2) chain-of-thought reasoning that requires looking up information from the in-context tokens as LLM’s generation continues.

To provide empirical support, we present two new sets of experiments.

**Results for two summarization tasks from LongBench**:
We have added results for two summarization datasets from LongBench (GovReport and QMSum; Table 10 in Section A.2 in the updated manuscript). For these two tasks, our method consistently outperforms SnapKV (the best baseline) for the two models we tested, especially for a smaller stride.  We have also added 11 other tasks from LongBench in Table 10 and Table 11.

**Experiment setting for a synthetic task that requires longer generation**
We further design a new synthetic task which requires the model to leverage various information in the context as the generation continues. We have added it to Section A.7 in the updated manuscript and describe the setting briefly in the reply below.

---

> ### Author Response · Authors · 2024-11-28
> **General Response (Further comparison with SnapKV) [2/2]**
>
> **Chain-of-key**:
> We define the synthetic task as “chain-of-key” and describe it below.
>
> The context consists of names of keys, each of which contains N number of words, for instance:
> ```apricot-waggish```.The model is tasked to generate a sequence which consists of a list of keys from the context, such that the first word of the next key is the last word of the current key. For example: ```waggish-fishery, fishery-mosquito, mosquito-perfume, perfume-panda, panda-juice, juice-willow, willow-bronco, bronco-creditor, creditor-bathhouse, bathhouse-woman``` Please refer to Table 13 in the updated manuscript for example input.
>
> Evaluation: We evaluate the length of the valid chain. A valid chain needs to satisfy two criteria: (a) the key must be in the context and (b) the first word of the current key must be the last word of the previous key. Please refer to Table 14 for example output and their scores.
>
> We report performance as well as decoding time for all methods. As our experiment shows that LLaMA-3.1-8B is unable to perform this task (accuracy of 0.11 with vanilla attention setting), we conduct an experiment with LLaMA-3.1-70B base model.
>
> **Results** We find that Recycled Attention consistently outperform baselines that evicts token from KV cache (SnapKV, StreamingLLM) as well as the StreamingLLM++ method, which perform full attention occasionally. SnapKV achieves an accuracy of 0.11, meaning that it is only able to generate a valid key for the first step. We also find that decreasing stride consistently improves performance for Recycled Attention. Please refer to Section A.7 in the Appendix for more details.
>
> |    | Method         | K    | S  | Accuracy | Time    |
> |----|----------------|------|----|----------|---------|
> | 1  | Vanilla        | -    | -  | 0.53     | 13.78   |
> | 2  | StreamingLLM   | 4096 | -  | 0.03     | 12.23   |
> | 3  | SnapKV         | 4096 | -  | 0.11     | _14.43_ |
> | 4  | StreamingLLM++ | 4096 | 20 | 0.03     | 12.41   |
> | 5  | Recycled       | 4096 | 20 | 0.14     | 12.88   |
> | 6  | StreamingLLM++ | 4096 | 15 | 0.03     | 12.47   |
> | 7  | Recycled       | 4096 | 15 | 0.17     | 13.21   |
> | 8  | StreamingLLM++ | 4096 | 10 | 0.04     | 12.61   |
> | 9  | Recycled       | 4096 | 10 | 0.19     | 13.69   |
> | 10 | StreamingLLM++ | 4096 | 5  | 0.06     | 12.82   |
> | 11 | Recycled       | 4096 | 5  | 0.38     | _15.20_ |

---

### Author Response · Authors · 2024-12-03
**General Response to AC and reviewers (rebuttal summary)**

Dear AC and reviewers,

Thank you for your service and for organizing and contributing to the review of our works. The review process has helped us greatly improve our manuscript.

We summarize the new results added to the manuscript in response to reviewers’ concerns and suggestions.

**A new section on dynamic stride:** Multiple reviewers (Reviewer h8fr, kJ4i) suggested dynamic scheduling with an adaptive stride S, instead of the fixed stride we experimented with in our initial submission. We have added a new section (Section 6) on dynamic stride based on query similarity. Our experiment shows that compared to a fixed stride (our implementation in the submitted draft), dynamic stride achieves similar performance with less decoding time (Table 6).

**More baselines:** We have added a new baseline (SnapKV), based on the suggestions from reviewers (Reviewer PFMm, NjEN, iC8W) to all experiment settings in our initial submission (RULER: Table 2 and 3; language modelling: Table 4), and the two sets of new experiments (LongBench: Table 10 and Table 11; Chain-of-keys: Table 12) for the two models we experimented with. Our method outperforms this new baseline, with competitive/better performance and a faster decoding speed. We have also added a discussion on the other baselines suggested by the reviewers in Section 7.

**More benchmarks:** As multiple reviewers suggested us to experiment on more datasets (Reviewer PFMm, NjEN, iC8W), we have added two sets of new experiments, including 11 datasets from LongBench (Table 10 and 11 in Section A.6) and a new synthetic dataset (chain-of-key, in Section A.7). Our method performs better than KV cache eviction methods (such as SnapKV) for longer generation which requires leveraging different information in the context based on what has been generated (two summarization datasets from GovReport and QMSum from LongBench, reported in Table 10; and the new “chain-of-key” task, reported in Table 12).

**Other experiments, analysis and manuscript improvements:** We have also added more results, analysis and discussion for other parts of the paper based on reviewers’ feedbacks:
* Ablation study of attention score aggregation method for GQA models (Section 3.3, Table 7; Reviewer iC8W)
* Ablation study of the choice of S (Table 8 and 9 in Section A.1; Reviewer kJ4i)
* Results on RULER for continued pre-training with Recycled Attention (Table 5 in Section 5; Reviewer PFMm)
* H2O's performance for attention overlap analysis in Figure 2 (Reviewer PFMm)

Together, we present a simple approach (Recycled Attention) to speed up LLM inference. Our method differs substantially from previously proposed KV cache eviction methods, which focus on memory efficiency. Instead, our method achieves speed-up by dynamically constructing a small KV cache for generation based on previous tokens’ attention patterns. We conduct comprehensive experiments (three tasks, 30+ datasets on two long-context models) and show that our method performs competitively or better than baseline methods. We include analysis ablating hyperparameters. We experiment with two extensions of Recycled Attention through (1) dynamic striding and (2) continued pre-training, both directions further improve the performance-efficiency tradeoff.

---

### Meta-Review · Area_Chair_pENc · 2024-12-18

**Metareview:**

This paper introduce a method for accelerating generation speed for long context lengths of LLMs by alternating between full context attentions and attention for an input token subet.

Although this paper addresses an important task, is well-written, and includes diverse experimental results, reviewers raised ciritical concerns such as lack of novelty and heuristic-based methods due to the existence of similar ideas in kv cache research.

AC also agrees with the concerns and think this paper is not sufficient for ICLR quality.

So, AC recommends rejecting this paper.

**Additional Comments On Reviewer Discussion:**

The initial scores were 6, 5, 1, 3, 5

The reviwers' main concerns include limited experiments (data and baselines), non-negligible performance degradation, and lack of novelty.

The authros tried to address the some issues, and some reviewers raised their scores, thus the final scores are 6, 5, 5, 5, and 6.

During AC-reviewer discussion, reviewers appreciated the authors' efforts. However, they agreed that the current contributions are not sufficient for ICLR quality.

---

### Decision · Program_Chairs · 2025-01-22

Reject